# FRIEDA: Benchmarking Multi-Step Cartographic Reasoning in Vision-Language Models

**Jiyoon Pyo**[1]   **Yuankun Jiao**[1]   **Dongwon Jung**[2]   **Zekun Li**[1]
**Leeje Jang**[1]   **Sofia Kirsanova**[1]   **Jina Kim**[1]   **Yijun Lin**[1]   **Qin Liu**[2]   **Junyi Xie**[1]
**Hadi Askari**[2]   **Nan Xu**[3]   **Muhao Chen**[2]   **Yao-Yi Chiang**[1]

[1]University of Minnesota-Twin Cities   [2]University of California, Davis
[3]University of Southern California

🤗 https://huggingface.co/datasets/knowledge-computing/FRIEDA
🔗 https://github.com/knowledge-computing/FRIEDA

## Abstract

Cartographic reasoning is the skill of interpreting geographic relationships by aligning legends, map scales, compass directions, map texts, and geometries across one or more map images. Although essential as a concrete cognitive capability and for critical tasks such as disaster response and urban planning, it remains largely unevaluated. Building on progress in chart and infographic understanding, recent large vision language model (LVLM) studies on map visual question-answering (VQA) often simplify maps as a special case of charts. In contrast, map VQA demands comprehension of layered symbology (e.g., symbols, geometries, and text labels) as well as spatial relations tied to orientation and distance that often span multiple maps and are not captured by chart-style evaluations. To address this gap, we introduce FRIEDA, a benchmark for testing complex open-ended cartographic reasoning in LVLMs. FRIEDA sources real map images from documents and reports in various domains (e.g., geology, urban planning, and environmental assessment) and geographical areas. Following classifications in Geographic Information System (GIS) literature, FRIEDA targets all three categories of spatial relations: *topological* (border, equal, intersect, within), *metric* (distance), and *directional* (orientation). All questions require multi-step inference, and many require cross-map grounding and reasoning. We evaluate eleven state-of-the-art LVLMs under two settings: (1) the *direct* setting, where we provide the maps relevant to the question, and (2) the *contextual* setting, where the model may have to identify the maps relevant to the question before reasoning. Even the strongest models, Gemini-2.5-Pro and GPT-5-Think, achieve only 38.20% and 37.20% accuracy, respectively, far below human performance of 84.87%. These results reveal a persistent gap in multi-step cartographic reasoning, positioning FRIEDA as a rigorous benchmark to drive progress on spatial intelligence in LVLMs.

## 1 Introduction

Recent advances in large vision-language models (LVLMs) have markedly improved multimodal reasoning, with strong results across diverse visual question-answering (VQA) tasks (Dong et al., 2024; Souibgui et al., 2025). Education and cognitive science research characterizes reasoning as a broad capability that spans numeric reasoning, logical deduction (Holyoak & Morrison, 2012), and textual interpretation (Wharton & Kintsch, 1991), as well as interpreting pictures (Mayer, 2020), spatial data (Li et al., 2025), and map images (Goodchild, 2012). Extensive LVLM benchmarks cover many of these facets: visual numeracy in chart and infographics (Lin et al., 2025; Mathew et al., 2022; Masry et al., 2022), document and layout reasoning (Duan et al., 2025; Mathew et al., 2021), multi-image inference (Kazemi et al., 2025; Xia et al., 2025), and even spatial relations in natural images (Shiri et al., 2024). However, reasoning over maps, also a core human compe-

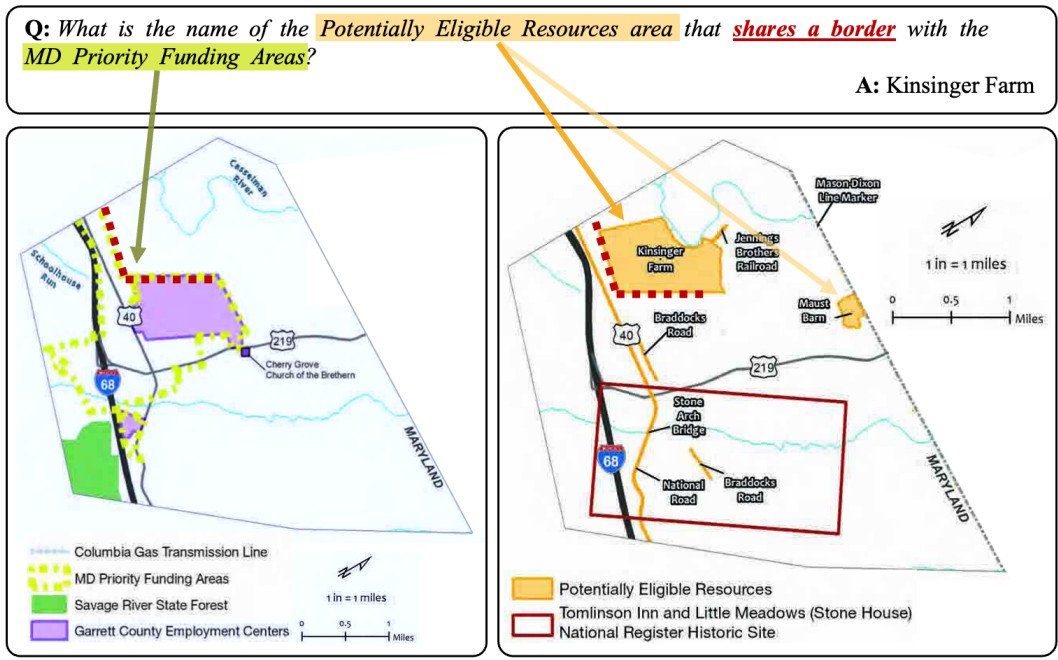

Figure 1: Example of a FRIEDA question requiring multi-map, multi-step cartographic reasoning. To solve the question, the model must (1) use each legend to locate the two referenced regions, (2) evaluate the *border* spatial relation between them, and (3) read the map label of the qualifying feature to answer "Kinsinger Farm."

tence (Tversky, 2003; Kastens & Ishikawa, 2006; Ishikawa & Newcombe, 2021), which we refer to as **cartographic reasoning**, remains under-examined in LVLMs.

Unlike natural images, maps encode information with an abstract, symbolic visual grammar (e.g., map scales, compass/north arrows, and thematic symbology) (Buckley, 2006), which demands a deeper interpretation than simple pattern recognition. Mastery of these elements must be coupled with the comprehension of spatial relations that are commonly grouped into topological reasoning (e.g., detecting shared boundaries), metric inference (e.g., converting map lengths to real-world distances through the map scale), and directional reasoning (using a compass or north arrow) (Clementini et al., 1993; Cohn & Renz, 2007). In addition, human map-reading competencies (Liben et al., 2010; Muir, 1985) frequently require these inferences across multiple maps within a single document. Correctly answering a map question, therefore, draws on map-specific skills (Hegarty & Waller, 2005) such as interpreting map elements, reasoning over spatial relations, and integrating evidence across multiple maps, as well as broader capabilities emphasized in LVLM research that include text grounding (Singh et al., 2019; Sidorov et al., 2020), numeric and logical inference (Lu et al., 2024a; Hu et al., 2023), multi-image integration (Wang et al., 2024a; Xia et al., 2025), and retrieval (Wang et al., 2025a; 2024b). A cartographic reasoning benchmark can therefore probe comprehensive reasoning and provide a clear understanding of the spatial intelligence of LVLMs.

A growing line of work began to evaluate LVLMs on map VQA, yet these benchmarks do not fully assess cartographic reasoning. Earlier datasets pose chart-style questions that can be answered without interpreting spatial relations, which bypasses the topological, metric, and directional inferences that are central to map comprehension (Koukouraki et al., 2025; Chang et al., 2022). Other efforts cover only a subset of relations as they target specific tasks such as navigation (Feng et al., 2025; Kazemi et al., 2025) or entity identification (Dihan et al., 2025). While suitable for those objectives, such coverage is insufficient for evaluating human-like map understanding (Liben et al., 2010). Many benchmarks also restrict the stylistic variability of maps. Some focuses on choropleths (Koukouraki et al., 2025; Chang et al., 2022; Mukhopadhyay et al., 2025), others rely on maps created with map-coloring tools (Srivastava et al., 2025) or common web basemaps (Kazemi et al., 2025; Dihan et al., 2025). Several others focus on limited thematic domains (e.g., geology (Huang

et al., 2025)) or on restricted geographic coverage (Chang et al., 2022; Srivastava et al., 2025). These constraints overlook the heterogeneity in toponyms, labeling conventions, projections, and symbology that real-world cartography demands (Slocum et al., 2022; Robinson et al., 1978). Multi-map reasoning is rarely evaluated, with limited exceptions (Kazemi et al., 2025), even though practical cases often require integrating evidence across multiple maps (e.g., reconciling transit maps with future land-use maps for urban planning) and aligning overlapping information (Lupien & Moreland, 1987). Moreover, although document-level multimodal understanding is emphasized in other LVLM benchmarks, existing map VQA benchmarks seldom require selecting the correct map among many images in long reports, despite government documents and technical documents containing numerous, visually similar, context-dependent maps (Federal Emergency Management Agency, 2025; U.S. Environmental Protection Agency, 2025; SEDAR+, 2025). As a result, current map VQA settings underestimate the demands of comprehensive map understanding, leaving it unclear whether LVLMs possess human-like map-reading competencies. Full cartographic reasoning remains beyond the scope of what existing VQA benchmarks assess.

We introduce FRIEDA, a benchmark designed for evaluating **multi-map, multi-step, comprehensive cartographic reasoning** in LVLMs. We curate maps from public documents of various thematic domains (e.g., geological surveys, planning reports, environmental studies) to develop questions that require models to interpret maps as they appear in reports, mirroring practical scenarios in which a reader must synthesize evidence from maps embedded in a document (see Figure 1). The collection encompasses a diverse range of styles, projections, and scales. We create each question such that it requires (1) reasoning over topological, metric, and directional relations, (2) interpreting map elements and their semantics (e.g., legends, map scales, and north arrows), and, when applicable, (3) integrating information across multiple maps, and (4) selecting the appropriate map(s) from a document to answer the query. To probe genuine reasoning rather than random guessing, the answers are in a free-form (not multiple-choice) format. The benchmark evaluation comprises two settings: a *direct* setting, in which the model is given the relevant map images along with the question to evaluate map comprehension, and a *contextual* setting, in which the model must first retrieve the correct maps from a broader within-document collection before answering. We score outputs using a unified, task-aware protocol aligned to the three spatial-relation categories. We evaluate textual responses (topological and semantic labels) with LLM-as-Judge (Gu et al., 2025), distance responses (numeric values with units) with unit-aware parsing and mean absolute percentage error (MAPE), and directional responses (cardinal directions for relative position) with angular tolerance over the eight directions. We compare the result against the human upper bound derived from multi-annotator agreement to contextualize LVLM performance. By aligning our tasks with the competencies expected of human map-readers (Goodchild, 2012; Liben et al., 2010) and explicitly targeting compositional cross-image inference that is largely absent from prior map VQA work, FRIEDA fills a crucial gap in state-of-the-art LVLM evaluation.

Across 11 LVLMs (both proprietary and open source), we find that even state-of-the-art models struggle with multi-step cartographic reasoning. With FRIEDA-direct, where the relevant maps are provided, the best-performing model (Gemini-2.5-Pro) correctly answers fewer than 40% of the questions, far below human performance ($> 80\%$). Overall accuracy remains essentially unchanged in the contextual setting, indicating that retrieval and disambiguation are not the primary bottlenecks; the core difficulty lies in cartographic reasoning itself. Our error analysis highlights recurring failures, such as misreading legends (confusing symbol shapes and colors) and misaligning information across maps when map styles, projections, or map scales differ. We also observe heterogeneous strengths across models (e.g., GPT-5-Think on multi-map questions and Claude-Sonnet-4 on distance queries). However, overall accuracy remains low, highlighting the gap between current LVLMs and the multi-step, cross-image cartographic reasoning skills required.

We organize the remainder of the paper as follows. Section 2 formalizes the tasks and core skills of cartographic reasoning; Section 3 describes the benchmark design and dataset statistics; Section 4 details the models, experimental setup, and evaluation protocol, and reports the results; Section 5 presents ablations and error analyses.

## 2    TASK DEFINITION

Cartographic reasoning is the ability to interpret maps and draw justified inferences from them. In FRIEDA, we design questions to assess core map-reading competence while mirroring realistic document use, where a reader may need to navigate a document to locate the relevant map(s). All questions require (1) reasoning over *spatial relations*, (2) interpreting heterogeneous *map elements*, and (3) integrating evidence across *multiple maps* when necessary. We also include a (4) *contextual setting* in which additional maps are provided, requiring the model to identify relevant map(s) before performing the reasoning. We detail these categories and the accompanying taxonomy below.

**Spatial Relation**    Spatial relations describe how geographic features relate in space (Carlson & Logan, 2001), how they are positioned in space (Majic et al., 2021), and how their geometries interact (Renzhong, 1998). In geographic information systems (GIS) and spatial cognition, these relations are often grouped into three categories: topological, metric, and directional (Cohn & Renz, 2007; Clementini et al., 1993). To make these abilities measurable and comparable, FRIEDA separates questions by spatial relation type and grounds the topological portion in the 9-intersection model (Clementini et al., 1993). We consolidate finer-grained subtypes into their broader categories (e.g., *cross* classified as *intersect*, and *contain* classified as *within*), yielding four topological classes: *border* (shared boundary between regions), *equal* (coincident geometries), *intersect* (crossing or overlap of features), and *within* (containment or inclusion of one area inside another). We complement these with one metric primitive, *distance*, and one directional primitive, *orientation*. Together, these six relations maintain the expressiveness of spatial queries while aligning with users' intuitive spatial reasoning.

**Map Elements**    Maps are symbolic representations that encode spatial information through abstract conventions (Slocum et al., 2022). Therefore, interpreting map elements is a distinct skill central to cartographic reasoning. The key elements we target are *map text* (place and feature names), *legends* (mappings from color, icons, and patterns to semantic classes), *map scales* (measurements that convert the map distance to the real-world distance), and the *compass* (ESRI, 2021). The styles of these components vary widely across maps: map texts may use different typography or placement rules (Monmonier, 2015), legends may use continuous color ramps or discrete pictograms (Slocum et al., 2022), map scales may appear as bars or frames around the map (Robinson, 1995), and the compass may be a compass rose or a north arrow (Slocum et al., 2022). Practical map interpretation requires grasping the concepts of map elements rather than simply recognizing their shapes. Consequently, our design includes questions that require reading map texts, decoding legends, using the map scale, and applying orientation to demonstrate true map literacy by linking abstract visual encodings to their underlying semantics.

**Multi-Map Reasoning**    Beyond interpreting spatial relations and map elements, practitioners regularly perform cross-map comparison and fusion to synthesize multiple map editions or thematic layers (Lupien & Moreland, 1987). Our multi-map setting reflects this practice: we curate questions that present two or more maps together and require the model to integrate evidence by aligning shared symbols, reconciling differences in labels, map scales, and orientation, and identifying co-referent regions or features (Foody, 2007). Extracting distributions and patterns is widely recognized as a core capability (Ishikawa, 2016; Rexigel et al., 2024; Morita & Fukuya, 2025). By testing this setting, we move beyond isolated spatial computation to evaluate deeper cartographic reasoning across varied depictions of the same space.

**Contextual Setting**    To mirror practical workflows (Mathew et al., 2021; Tanaka et al., 2023), we evaluate a contextual setting (FRIEDA-contextual) in which a model must identify the relevant map before answering a question. In this scenario, we provide the model with multiple maps from the same source (i.e., a document), and the model must perform within-document retrieval using cues in the maps, such as titles, legends, or labels. By evaluating model performance on FRIEDA-contextual, we capture a core aspect of real map use: the model must understand how map elements encode meaning and leverage that understanding to select the required map from thematically related alternatives that vary in data layers, geographic extent, or purpose (Ishikawa, 2016).

| Statistics | Number |
|---|---|
| Total questions | 500 |
| Total number of documents | 210 |
| Total number of images | 17,030 |
| Map text | 366 (73.2%) |
| Legend | 417 (83.4%) |
| Compass | 137 (27.4%) |
| Scale | 46 (9.2%) |
| Single-map | 202 (40.4%) |
| Multi-map | 298 (59.6%) |
| Avg maps in contextual | 9.5 |
| Relevant:Irrelevant | 1:5.71 |

Table 1: Key statistics.

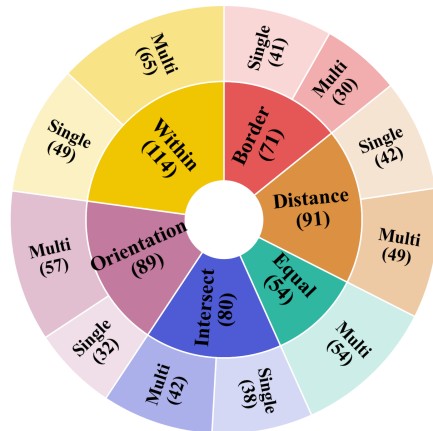

Figure 2: Question distribution by spatial relation (inner) and map count (outer). Sizes are proportional to the number of questions in each category.

# 3 FRIEDA

We present FRIEDA, a benchmark for assessing LVLM's comprehensive cartographic reasoning, with an emphasis on cross-map (i.e., multi-image) scenarios. This section summarizes the benchmark statistics and details the dataset curation procedure.

## 3.1 BENCHMARK STATISTICS

Table 1 shows that FRIEDA comprises 17,030 map images drawn from 210 documents. To capture real-world variability, these maps span diverse geographies (32 countries) and six thematic domains, exhibiting heterogeneous styles, including varied color palettes, legends, and symbol conventions. The benchmark comprises 500 questions, of which 202 are single-map and 298 are multi-map. The multi-map subset consists primarily of two-map comparisons (295 questions), with a few cases requiring reasoning across three (2 questions) or four (1 question) maps.

Figure 2 reports the question distribution for each spatial relation, stratified by single- and multi-map questions.[1] The distribution is roughly balanced across relations and settings. We provide the detailed counts by spatial relation and setting in Appendix C.1 and include a representative example of each in Appendix D.

For FRIEDA-contextual, we provide between 2 and 9 irrelevant maps in addition to the relevant map(s) required to answer the question. The total number of input images averages 9.54 maps ($\sigma = 1.27$) per question across both the single- and multi-map settings, with an average relevant-to-irrelevant maps ratio of 1:5.71.

## 3.2 BENCHMARK CONSTRUCTION

This section describes the construction of FRIEDA, which proceeds in four stages: map image collection, question generation, pre-annotation curation, and validity verification.

**Map Image Collection**   To capture stylistic and geographic diversity, we curate maps from publicly available government and multilateral reports across domains including geology (SEDAR+, 2025), national park management (National Park Service, 2025), environmental assessments (U.S. Environmental Protection Agency, 2025; Environmental Protection Agency, 2025; Ministry of Agriculture Climate Change and Environment, 2025), disaster response (Federal Emergency Management Agency, 2025), urban planning (Seattle Planning and Community Development, 2024; De-

---

[1]As *equal* denotes identical geometry (i.e., identical location and size), the benchmark contains no single-map *equal* questions.

partment of Community Development, 2025; Urban Redevelopment Authority, 2025; City of Cape Town, 2025), and infrastructure investment (AIIB, 2025). We limit the sources to documents written in Latin characters to focus on cartographic reasoning rather than translation. We extract images using Idefics3-8B (Laurençon et al., 2024) with a custom prompt (Appendix B.1) and manually verify that each extracted set contains only cartographic maps (examples of excluded non-maps appear in Appendix B.1.1). To support FRIEDA-contextual, we retain only documents with at least four maps. We select contextual maps based on their page proximity to the target map; this ensures they are thematically and stylistically related to the target maps. We then shuffle the map order to prevent LVLM from using positional cues to identify the target maps.

**Question Generation**  For each collected map, we use GPT-4 and GPT-o3 (see Appendix B.6) to propose candidate questions, their corresponding targeted spatial relations, and reference answers. Rather than enforcing fixed templates, we accept any phrasing that unambiguously expresses the intended relation to reflect various forms of paraphrases (e.g., "Is A within B?" vs. "Does B contain A?"). To ensure the benchmark tests visual cartographic reasoning over memorization or web retrieval, we screen each question for searchability using GPT with web search enabled. In addition, we discard questions that can be answered without inspecting the map image. We label each question's answer type as textual (short span or categorical label), distance (numeric with units), or direction to enable analysis by response modality.

**Pre-Annotation Curation**  All LLM-proposed candidate questions undergo a pre-annotation curation stage. The two question curators (one with 7 years of GIS experience and another with 2 years of experience in geospatial data) manually verify gold answers against source maps and rewrite or discard ambiguous questions. This step ensures that FRIEDA consists solely of clear, high-quality questions prior to the broader validation phase with the annotator.

**Annotation Pipeline**  We validate each question using annotations from 11 Ph.D. researchers (eight with map expertise), collected over four weeks. Annotators confirm that each question is answerable from the provided map(s) and, for multi-map questions, verify that all maps are required to answer the question. To prevent bias, curators do not validate their own edits. We retain the question only if the majority ($\geq 2/3$) agrees with the gold answer. In a rare case (currently two questions in FRIEDA) where all three annotators agree on an answer that contradicts the gold answer, we conduct a secondary review to update the gold answer if consensus is reached. In total, we remove 61 questions that do not meet the agreement threshold of $\geq 2/3$. Appendix B.3 details the instruction prompt provided to the annotators, and Appendix B.5 shows the annotation interface.

## 4 EXPERIMENTS

This section details the experimental setup, baselines, and evaluation metrics, and then presents the main result, showing that FRIEDA is a challenging benchmark even for the strongest LVLMs.

### 4.1 EXPERIMENTAL SETUP

**Models**  We evaluate 11 LVLMs with multi-image support on FRIEDA. For proprietary models, we test three models: Gemini-2.5-Pro (Gemini Team, 2025), GPT-5-Think (OpenAI, 2025), and Claude-Sonnet-4 (Anthropic, 2025). For open source models, we consider eight model families and evaluate the largest available model from each family: LLaVA-NeXT-110B (Li et al., 2024b), GLM4.5V-108B (Team et al., 2025), InternVL3-78B (Chen et al., 2024b), LLaVA-OneVision-72B (Li et al., 2024a), Qwen2.5VL-72B (Bai et al., 2025), InternVL3.5-38B (Wang et al., 2025b),[2] Ovis2-34B (Lu et al., 2024b), and Ovis2.5-9B (Lu et al., 2025).

To enforce determinism in open-source models, we set `do_sample=False` and `temperature=0`. For proprietary models, we use the default settings for each model with maximum reasoning enabled (e.g., `reasoning=high` for GPT-5-Think) and append the instruction "Do not use search" to disable external retrieval. All models receive the same set of instructions that human annotators receive (Appendix B.4).

---

[2]We evaluate the 38B variant rather than the 241BA28B variant as the latter activates only 28B parameters during inference. We report the results for the 241BA28B setting in Appendix F.2.

**Evaluation metrics**   Answers in FRIEDA fall into three categories: textual, distance, and direction. For textual answers, we employ an LLM-as-Judge (Gu et al., 2025) method, utilizing Mistral Small 3.1 (Mistral AI, 2024) as the evaluator.[3] The full judge prompt appears in Appendix E.1. This setup handles minor variation (e.g., 'Cypress Creek' vs. 'Cypress') by matching semantics rather than identifying exact string equality. For distance-based answer, we report mean absolute percentage error (MAPE) and consider predictions within 20% error as correct, following Lewis (1982). For directional answers, we mark a response correct if it matches the target cardinal direction within one adjacent label (e.g., if the gold answer is North, accept North, North West, and North East), reflecting the perceptual nature of the labels. We validate the reliability of the evaluation method against manual annotations, achieving a Cohen's $\kappa$ of 0.9028 across all judged questions, which supports its suitability for evaluation.

## 4.2 EVALUATION RESULTS

Figure 3 summarizes the overall performance, and Table 2 reports accuracy by spatial relation. As FRIEDA retains questions with at least 2/3 annotator agreeing on the gold answer, we report accuracy for two subsets: *All-Agree*, where all three annotators agreed, and *Partial-Agree*, where 2/3 annotators agreed. *All-Agree* items serve as an indirect indicator of questions that are easier and less ambiguous for the annotators under our task and instructions, whereas *Partial-Agree* items may be considered as intrinsically more difficult or ambiguous to answer correctly. We also report the *Overall Accuracy*, which aggregates both subsets. Even the strongest LVLM (Gemini-2.5-Pro) remains below 40% overall accuracy, well behind human performance at 84%. The best open-source result (Ovis2.5-9B-Think) achieves 24% overall accuracy, underperforming proprietary systems and falling far below human performance. We find no clear relationship between model size and performance, suggesting that training data, training objectives, and explicit reasoning mechanisms matter more than scale for cartographic reasoning. We present further analyses in the following section and in Appendix F.2.

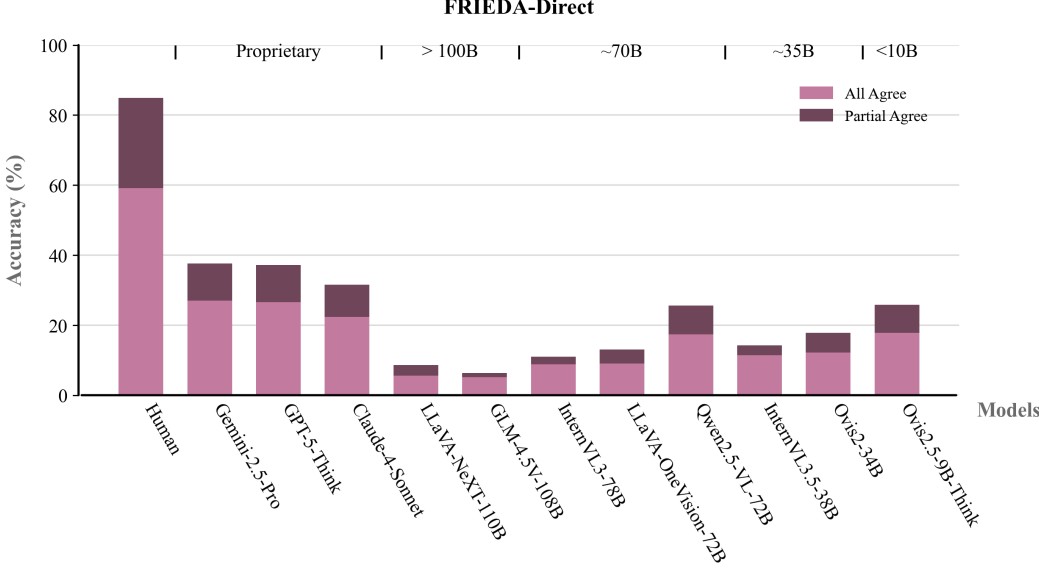

Figure 3: Overall accuracy of different models on the FRIEDA-direct benchmark.

## 5 ANALYSIS

**Error analysis on Gemini Pro**   To pinpoint where LVLMs fail, we analyze Gemini-2.5-Pro on the *All-Agree* subset (167 questions in total). This ensures that our analysis targets distinct model

---

[3]Mistral is not the language backbone of any tested LVLM, thereby reducing potential bias ( Panickssery et al. (2024)).

| | Overall (500) | Border (71) | Distance (91) | Equal (54) | Intersect (80) | Orientation (89) | Within (115) |
|---|---|---|---|---|---|---|---|
| Human Average | **84.87** | **89.00** | **78.28** | **89.10** | **85.53** | **91.80** | **88.08** |
| *Proprietary LVLMs* | | | | | | | |
| Gemini-2.5-Pro | **38.20** | 32.39 | 25.27 | 33.33 | 28.75 | **71.59** | **35.34** |
| GPT-5-Think | 37.20 | 25.35 | **27.47** | **44.44** | **31.25** | 69.32 | 28.45 |
| Claude-Sonnet-4 | 31.60 | **33.80** | 23.08 | 37.04 | 22.50 | 56.82 | 21.55 |
| *Open Source LVLMs* | | | | | | | |
| LLaVA-NeXT-110B | 8.60 | 4.23 | 10.99 | 11.11 | 16.25 | 0.00 | 9.48 |
| GLM-4.5V-108B | 6.40 | 5.41 | 2.15 | 21.57 | 6.17 | 1.16 | 7.83 |
| InternVL3-78B | 11.00 | 1.41 | 4.40 | 12.96 | 5.00 | 34.09 | 7.76 |
| LLaVA-OneVision-72B | 13.00 | 9.86 | 10.99 | 5.56 | 8.75 | 29.55 | 10.34 |
| Qwen2.5-VL-72B | 25.60 | 11.27 | 14.29 | 25.93 | 17.50 | 55.68 | 25.86 |
| InternVL3.5-38B | 14.20 | 11.27 | 8.79 | 14.81 | 2.50 | 36.36 | 11.21 |
| Ovis2-34B | 17.80 | 25.35 | 13.19 | 25.93 | 26.25 | 2.27 | 18.97 |
| Ovis2.5-9B-Think | 25.80 | 12.68 | 20.88 | 24.07 | 22.50 | 51.14 | 21.55 |

Table 2: Overall and per spatial relation accuracy of human and LVLMs on FRIEDA-direct.

failures on questions that humans find straightforward. We assign each incorrect answer to a single primary error category. When multiple issues co-occur, we prioritize errors that occur earlier in the reasoning pipeline that propagate to downstream steps. The largest source of error is the misinterpretation of legends (25.61%): cases in which the model assigns colors or symbols to the wrong class. The remaining 23.78% is due to cross-map interpretation failures, which reflect difficulties in aligning the map scales and shared features across maps, and 16.46% is due to spatial-relation semantics error, which arises when the model mixes up spatial relations (e.g., labeling region B *within* A when it only *touches* A at the boundary). Map-element misunderstandings include mistakes with the map scale (9.76%; unit or ratio errors), map text (8.93%; selecting the wrong place or feature from labels), geometry or shape reference (3.66%; pointing to the wrong area on the map), and orientation (3.05%; ignoring a tilted compass). Finally, we observe generic VQA errors not specific to cartography, such as miscounting (6.71%), subject-object confusion (1.82%; referring 'A relative to B' as 'B relative to A'), and hallucination (1.20%). For the top three error categories, we provide examples and rationales returned by the three proprietary models in Appendix F.1.

**Performance by spatial relation** Figure 4 reports per-spatial relation accuracy for human annotators and the three proprietary models. LVLM performance broadly tracks the human baseline: both are most accurate on orientation and struggle most with distance. On questions where an annotator answers incorrectly, LVLMs are also incorrect 84.53% of the time. While GPT-5-Think and Gemini-2.5-Pro achieve comparable overall accuracy, GPT-5-Think is stronger on tasks that require multi-map reasoning (Table 10), indicating better integration of evidence across maps. This is most evident in the equal relation questions, a multi-map exclusive task, where GPT-5-Think's accuracy is nearly 13% higher compared to Gemini-2.5-Pro. Notably, Claude-Sonnet-4 is the strongest on distance questions, particularly those that require interpreting the map scale to compute exact distances.

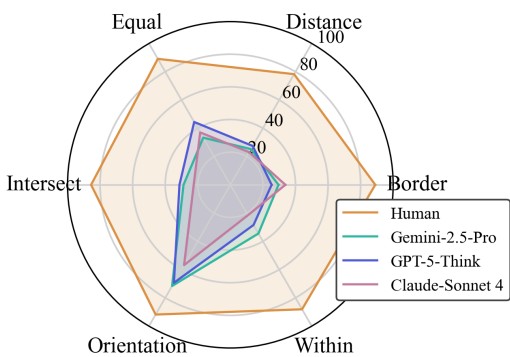

Figure 4: Per spatial relation accuracy (%) of human annotators and three proprietary LVLMs (Gemini-2.5-Pro, GPT-5-Think, and Claude-Sonnet-4) on FRIEDA-direct.

**Performance on contextual setting**   We observe a minimal difference in accuracy between the FRIEDA-direct (Figure 3 and Table 2) and FRIEDA-contextual (Appendix E.3). To verify that this is not an artifact of the accuracy metric, we directly compare the per-question performance of the eight open-source models under deterministic settings (i.e., `do_sample=False` and `temperature=0`). We observe 88.03% agreement in per-question performance between the direct and contextual settings, indicating that contextual images (maps from the same document that are not required to answer the question) rarely affect the model's predictions.

| Model | Accuracy (%) |
|---|---|
| Ovis2.5-9B | 19.00 |
| Ovis2.5-9B-Think | **24.80** |

Table 3: Performance of Ovis2.5 model on FRIEDA-direct.

**Impact of reasoning (think) on cartographic question types**   Despite being the smallest model tested with FRIEDA, Ovis2.5-9B-Think attains strong results (4th overall and 1st among open source models). To identify what drives this performance, we further evaluate Ovis2.5-9B with explicit reasoning (i.e., *Think*) disabled (Table 3). The overall accuracy of Ovis2.5-9B remains above the open-source average, indicating that model characteristics (e.g., architecture, training data) contribute to its strong performance. Enabling *Think* adds an additional 5% performance gain. To identify which question types benefit from explicit reasoning and whether it improves cartographic performance, we manually analyze the 60 questions that only the *Think* variant answers correctly. Reasoning helps mostly with cardinal-direction questions, where north faces the top of the image (48.33%), followed by multi-map alignment (23.33%). Additional improvements arise from correctly reading map text (15%), interpreting the map scale (5%), associating the legend with the symbol (5%), and counting (3%). Together, these patterns suggest that explicit reasoning primarily strengthens orientation-related and multi-map questions, which are central to cartographic reasoning, whereas yielding smaller gains in symbol and map-scale interpretation.[4]

## 6   RELATED WORK

**Document & Infographic/Chart VQA**   Recent benchmarks established baselines for LVLM reasoning over documents and designed graphics. In the document domain, DocVQA (Mathew et al., 2021) and DocVXQA (Souibgui et al., 2025) introduce a large-scale question-answering (QA) dataset over real forms and reports, while DocoPilot (Duan et al., 2025) extends evaluation to scientific articles, which involve embedded figures. For graphics, InfographicsVQA (Mathew et al., 2022) and InfoChartQA (Lin et al., 2025) test reasoning over rich layouts and charts. In general, frontier LVLMs reveal competence at high-level patterns, such as trends and extrema, but struggle with precise value extraction and robustness. FRIEDA evaluates these shortcomings in a cartographic setting where layout, symbols, legends, scales, and compass orientation interact tightly to measure how well LVLMs integrate these signals to answer map-based questions.

**Map VQA and Spatial Reasoning**   While recent map VQA benchmarks have expanded the scope of evaluation, they remain constrained to single-map tasks or specific domains. MapQA (Chang et al., 2022) evaluates question answering on choropleth maps and shows that general VQA and ChartVQA systems underperform on maps. MapWise (Mukhopadhyay et al., 2025) broadens the geographic coverage, while MapIQ (Srivastava et al., 2025) extends the map type coverage to include cartograms and proportional-symbol maps. MapEval (Dihan et al., 2025) assesses geospatial reasoning across various cities, and it reports consistent human-LVLMs performance gaps. Domain-specific efforts include PEACE (Huang et al., 2025) for geology, CartoMark (Zhou et al., 2024) for text extraction and recognition, and MapBench (Xing et al., 2025) and ReasonMap (Feng et al., 2025) for navigation. However, these benchmarks rarely test cross-image reasoning on heterogeneous sources and often rely on a limited set of spatial relations. While ReMI (Kazemi et al., 2025) explores the cross-image setting, the questions lack cartographic focus. We detail key differences between prior map VQA benchmarks and FRIEDA in Appendix G.

Spatial reasoning benchmarks, such as SpatialVLM (Chen et al., 2024a) and SpatialRGPT (Cheng et al., 2024), have advanced spatial perception and reasoning on natural images. In the geospatial

---

[4]We further evaluate the association between performance and model size in Appendix F.2

domain, GeoChain (Yerramilli et al., 2025) enhances tasks like geolocalization by inducing step-by-step geographic reasoning. However, these works do not engage with symbolic conventions unique to maps (i.e., legends, scales, compasses, and map texts). In contrast, our benchmark closes this gap by evaluating multi-step cartographic reasoning over heterogeneous, real-document maps, which requires models to integrate evidence across multiple figures and align legends, scales, and orientation to infer key spatial relations (i.e., border, distance, equal, intersect, orientation, and within).

## 7  CONCLUSION

We present FRIEDA, a benchmark for evaluating multi-step cartographic reasoning over real-world maps across six core spatial relations, where answering often requires aligning evidence across multiple maps. Our evaluation of 11 state-of-the-art LVLMs reveals a substantial gap between current performance and the proficiency required for reliable map understanding. Models frequently fail to ground symbologic map entities, to integrate evidence distributed across multiple maps, and to correctly interpret spatial topology. Our analysis further shows that these failures are not merely amplified versions of errors observed in prior VQA benchmarks, but also expose additional map-specific challenges, including symbol-to-semantics mapping, adherence to cartographic conventions, and geometry-aware reasoning. These findings highlight the need for novel architectures and effective training strategies that incorporate cartographic priors and enable explicit reasoning over map elements and spatial structure. To catalyze progress, we release FRIEDA along with the question taxonomy, baseline results, and evaluation scripts to enable fine-grained diagnosis and reproducible comparisons. We encourage the community to build on FRIEDA with methods that explicitly integrate text, symbology, and geospatial structure, moving toward LVLMs that reason robustly and reliably over real-world maps.

### ETHICS STATEMENT

We introduce a benchmark for evaluating cartographic reasoning in large vision-language models. We curate maps from publicly available documents (e.g., government reports, planning, and environmental studies) and retain only the figures necessary for research purposes. To the best of our knowledge, we use all materials under terms that permit research and non-commercial distribution.

All annotators provided informed consent. We collected no personal data about annotators beyond task performance. Our institution's IRB reviewed the annotation protocol and determined that the project does not constitute human subjects research; therefore, no further IRB review was required.

The benchmark inevitably reflects the patterns in the source documents and may exhibit representation bias, including uneven geographic coverage and map types, English-language focus, and unequal representation across regions and themes. We document these limitations and their potential impact in the dataset card (Appendix A) to aid transparency and interpretation.

### REPRODUCIBILITY STATEMENT

We released the benchmark dataset (images, QA JSON, and taxonomically classified questions) on HuggingFace, along with all prompt and configuration files used for annotation and inference. We also provide the full codebase for data loading, inference, and evaluation on GitHub, as well as code to replicate the annotation interface. The results obtained during testing are released at result branch.

### ACKNOWLEDGEMENTS

This material is based upon work supported in part by the National Science Foundation under Grant No. BCS-2419334. A subset of the authors received support from NSF Grants OAC-2531126 and ITE-2333736, as well as from the Amazon Nova Trusted AI Prize.

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

# A   DATACARD

We adopt the data statement framework of Bender & Friedman (2018) and integrate complementary fields from Datasheets for Datasets (Gebru et al., 2021) to centralize key information for the analysis, reuse, and deployment of FRIEDA.

## A.1   CURATION RATIONALE

We design FRIEDA to evaluate carographic reasoning (the ability to interpret map-specific symbols, comprehend spatial relations, and integrate evidence across one or more maps). We source questions from public documents to reflect map-reading tasks encountered in practice (e.g., planning, hazard assessment, and geology). High-level goals, task definitions, and design choices appear in the main text (Section 2 and Section 3). We further expand on the benchmark curation process in Appendix B.

## A.2   BENCHMARK COMPOSITION

- **Total size**: 500 validated questions; each question with 1 gold answer
- **Agreement**: Each question is annotated by three annotators; we record the problem-level agreement and mark items with unanimous agreement on the gold answer as *All-Agree*, and those with 2/3 agreement as *Partial-Agree*.
- **Modalities**: Every question involves one or multiple map image(s) and associated question text.
- **Spatial relations (6)**: Border, Equal, Intersect, Within, Distance, Orientation
- **Answer types (3)**: Textual (short text), Distance, and Direction
- **Provenance**: Public documents from 32 countries across six continents. Documents are from six domains (urban planning, environmental assessment, national park management, geologic reports, disaster and hazard reports, infrastructure and investment reports). Sources are detailed further in Section 3 and Appendix C.3.
- **Languages**: Questions and instructions are in English (en-US); source maps primarily use English labels but may include other languages written in the Latin script.

## A.3   DATA COLLECTION PROCESS

- **Acquisition**: We collected maps from public reports, then filtered for reading map elements and task suitability.
- **Question creation**: Curators wrote questions that required reading the legend, scale, and compass, and reasoning over one or more spatial relations; questions were rejected if (1) they were solvable without using any maps or (2) if question ambiguity could not be resolved by manual editing.

## A.4   ANNOTATOR DEMOGRAPHIC

We share the annotator demographics to contextualize potential biases while preventing re-identification.

- **Count**: 11 Annotators in total (2 also served as question curator)
- **Academic background**: Ph.D. Researchers [100%]
- **GIS/cartography background**: ≤1 year: [27%]; 1–3 years: [27%]; 3–5 years: [18%]; 5+ years: [27%].
- **Language**: All authoring and communication used American English (en-US). As the task focuses on cartographic symbols and spatial relations (not dialect), we do not report individual annotator nationalities. Non-native participation may introduce minor phrasing variance. We standardized qustion phrasing during review and removed questions flagged as ambiguous by ≥ 2/3 annotators.

### A.5 EVALUATION & METRICS

- **Primary metric**: Accuracy
- **Textual (LLM-as-Judge)**: After attempting exact string match, we use an LLM-as-Judge to compare model outputs to gold answers. Appendix E.1 provides the judging prompt for reproducibility.
- **Distance (MAPE)**: We apply mean absolute error (MAPE) and unit-aware parsing and consider all distance answers with in 20% as correct.
- **Direction**: We canonicalize directional answers to the eight cardinal directions and consider all cardinal direction within one adjacent unit as correct.

### A.6 KNOWN LIMITATIONS & BIASES

- **Regional representation bias**: As FRIEDA uses only English-language documents, regions where English is a dominant language are overrepresented, and non-English conventions and locales are not covered.
- **Domain skew**: The corpus emphasizes planning, environmental, and government reports with less coverage on other types of maps, such as nautical or military charts.

## B DETAILED BENCHMARK CONSTRUCTION

### B.1 MAP IMAGE FILTERING

We use Idefics3-8B (Laurençon et al., 2024) to filter map images from the document. To produce a strict Yes/No decision, we prompt the model:
```
Is this a cartographic map?  Answer only with Yes or No.
```
We consider any image for which the model responds `Yes` as a candidate map.

### B.1.1 NON-MAP EXAMPLES

We manually verify all map candidate images and remove those that we do not consider as maps. For example, although Figure 5 shows a silhouette of a city with subdivision, we consider it as a stylized graphic rather than a cartographic map. The image lacks essential map elements (i.e., map texts, legend, scale, and compass), which are needed to support cartographic reasoning. Without these components, we cannot reason about locations, distances, or spatial relationships; therefore, we exclude such images from our dataset and do not treat them as maps for FRIEDA.

### B.2 DEFINITION OF SPATIAL RELATION

Figure 6 visualizes the four topological spatial relations evaluated in FRIEDA: border, equal, intersect, and within.

### B.3 ANNOTATOR PROMPT

To standardize responses and minimize ambiguity, we supply annotators with a fixed instruction set (Figure 7). We introduce these guidelines during task onboarding and repeat them at the start of every question to promote a consistent answer format.

### B.4 LVLM SYSTEM PROMPT

To ensure consistency, we use the same instruction set provided to human annotators as the prompt for the LVLM system. As some LVLMs produce intermediate reasoning, we append a final line to standardize the output: `Give the final answer in 'Final answer: ⟨your answer⟩`. For the proprietary models, we additionally include the clause `Do not use online search` to prevent external browsing.[5]

---

[5]We add this clause as a precautionary measure; during the dataset construction phase, we verify that questions are not directly answerable through web search.

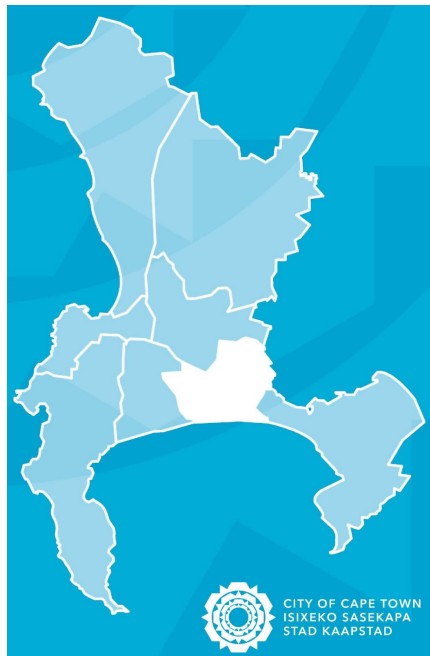

Figure 5: An example of a non-map image flagged by Idefics3-8B as a candidate map. The image is a graphic from the cover page of the document. We exclude it from the benchmark after manual verification, as we consider it a graphical image rather than a cartographic map.

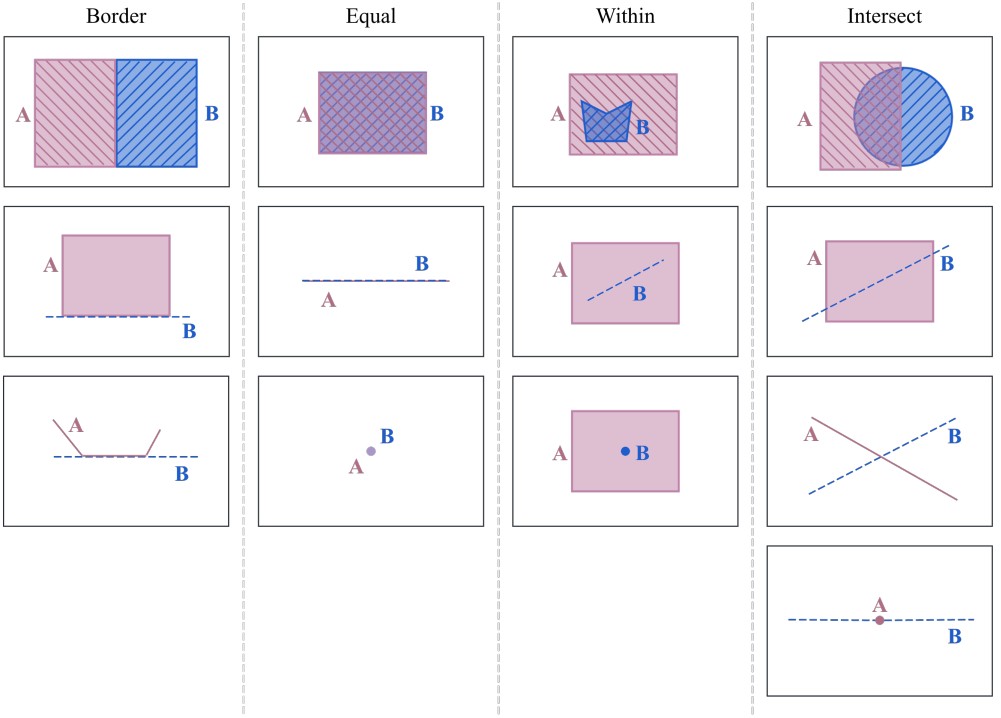

Figure 6: Illustrations of the spatial relations evaluated in the benchmark. Columns show *Border*, *Equal*, *Within*, and *Intersect*; rows provide representative cases across geometry types—areas, lines, and points.

For each one, please verify whether it can be answered (Q# Validation) using the provided map(s). If an image appears too small, click on the image. For question with multiple images, please mark whether all images were required to correctly answer the question (Q# M). You may use tools like a ruler or calculator, but do not use online search.

For each questions:
General:
- If question can be answered, write answer in short answer box
- If answer is a text from the map, copy it as it appears

Numerical Answers:
- Include units as indicated on the map *(Don't convert 1200m to 1.2km)*
- If both map frame and ruler scale is available, use the ruler scale
- If question asks for an area, use {unit}^2
- Use numerical values *(e.g., 4 instead of four)*

Directional Answers:
- Use 8 cardinal directions only: North, North East, East, South East, South, South West, West, North West
- Write 'North' or 'South' before 'East' or 'West'
- Notice that the north arrow compass do not always point upward

Multi-Part Answers:
- Separate with semicolon (;) *(e.g., Zone A; Zone B)*

Figure 7: Instruction provided to annotators at the beginning of every question.

```
Answer the questions based on the following criteria:
General:
* If question can be answered, write answer in short answer box
* If answer is a text from the map, copy it as it appears

Numerical Answers:
* Include units as indicated on the map (Don't convert 1200m to 1.2km)
* If both map frame and ruler scale is available, use the ruler scale
* If question asks for an area, use {unit}^2
* Use numerical values (e.g., 4 instead of four)

Directional Answers:
* Use 8 cardinal directions only:  North, North East, East, South East, South, South
West, West, North West
* Write 'North' or 'South' before 'East' or 'West'
* Notice that the north arrow compass do not always point upward

Multi-Part Answers:
* Separate with semicolon (;) (e.g., Zone A; Zone B)
Give the final answer in 'Final answer:  ⟨your answer⟩'
{Do not use online search}
```

Figure 8: System prompt used for LVLM inference. For readability in the figure, newline characters (\n) are shown as actual line breaks.

## B.5 ANNOTATION PLATFORM

We built a web interface (Figure 9) to collect annotator responses. We provide the annotator instruction (Figure 7) at the top of every question similar to how LVLMs receives the system instruction for each question. For each question, annotators see the question and its associated map image(s), then (1) enter a short free-text answer if it is considered answerable, (2) mark answerability by selecting either "Can be answered" or "Map doesn't contain information to answer the question" (the latter requires a brief justification), and (3) for multi-map questions, indicate whether all images are necessary to precisely answer the question without guessing.

## B.6 LLM TO GENERATE QUESTIONS

We use GPT-4 and GPT-o3[6] with a tailored prompt (Figure 10) to draft candidate questions for FRIEDA. In addition to the prompt, we supply 10 randomly selected map images for question generation. Two of the authors then manually review each candidate question, editing or discarding questions as needed to ensure correctness, clarity, and coverage of targeted spatial relations before adding them to the benchmark.

## C EXTENDED BENCHMARK DETAILS

This section provides expanded details on the distribution of questions within FRIEDA. To visualize the hierarchical nature of the task dimensions formalized in Section 2, we present a Sankey diagram (Figure 11). Additionally, we provide granular breakdown counts for other dataset attributes, including question frequency per spatial relation, national representation, and domain diversity.

## C.1 QUESTION COUNT PER SPATIAL RELATION

In Table 4, we report the number of questions in FRIEDA by spatial relation, including totals as well as the counts split into single-map and multi-map questions. The distribution is roughly balanced: *Within* is the largest class (23.0%), while *Equal* is the smallest (10.8%).

| Spatial Relation | Total Q Count | Single-map Q Count | Multi-map Q Count |
|---|---|---|---|
| Border | 71 (14.2%) | 41 (8.2%) | 30 (6.0%) |
| Distance | 91 (18.2%) | 42 (8.4%) | 49 (9.8%) |
| Equal | 54 (10.8%) | 0 (0.0%) | 54 (10.8%) |
| Intersect | 80 (16%) | 38 (7.6%) | 42 (8.4%) |
| Orientation | 89 (17.8%) | 32 (6.4%) | 57 (11.4%) |
| Within | 115 (23.0%) | 49 (9.8%) | 65 (13.0%) |

Table 4: Question statistics in FRIEDA across six spatial relations. The table reports the total number of questions per relation, along with their breakdown into multi-map and single-map settings.

## C.2 EXAMPLE QUESTION PER SPATIAL RELATION

In Table 5, we present one sample question for each spatial relation, split by map count (single-map vs. multi-map).

## C.3 NATION AND DOMAIN COVERAGES

**Nation Coverage** FRIEDA includes maps from government documents and multilateral reports from 32 countries across six continents (Figure 12; Table 6). We also report the ten most-represented countries by question count in Figure 13.

---

[6]Questions are generated before the release of GPT-5.

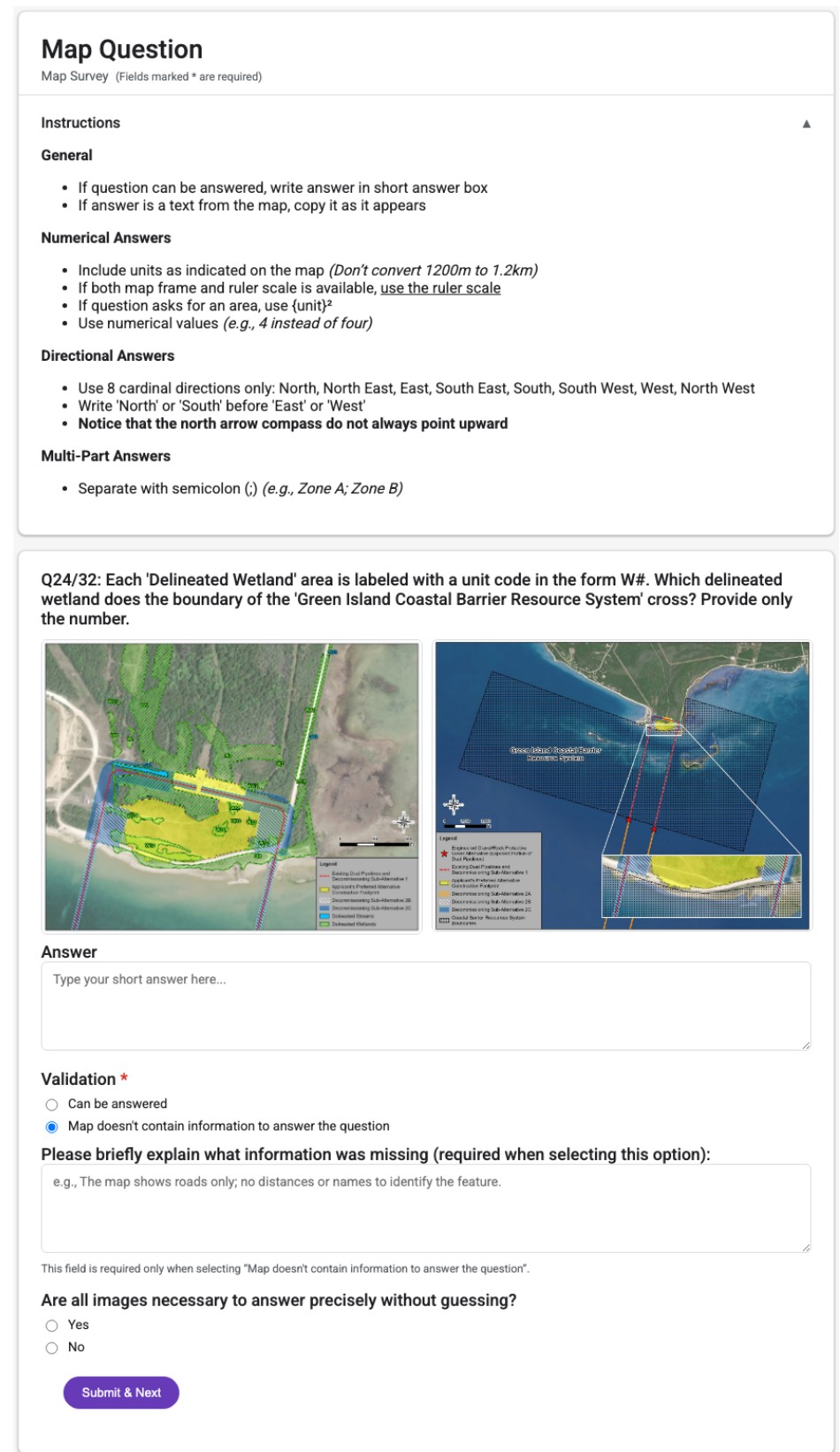

Figure 9: Annotation interface for validating questions of FRIEDA.

I'm trying to create a benchmark dataset to test out generative AI's ability on complex cartographical reasoning on maps. The hard questions we should provide in this benchmark should leverage information from one or a few of the given maps above, and should involve some reasoning. Also, the questions should follow these criteria:

- Answer should be self-contained, non-binary, and not-multiple choice questions.
- Question should not be solved by searching online - We assume that the image to refer to is not known when answering the question.
- We assume that the image to refer to is not known when answering the question.

Give a set of questions, the maps to refer to, and the answer to the question. Target spatial relation is {Spatial Relation}.

Figure 10: Question-generation prompt used to prompt for candidate questions to either GPT-4 or GPT-o3.

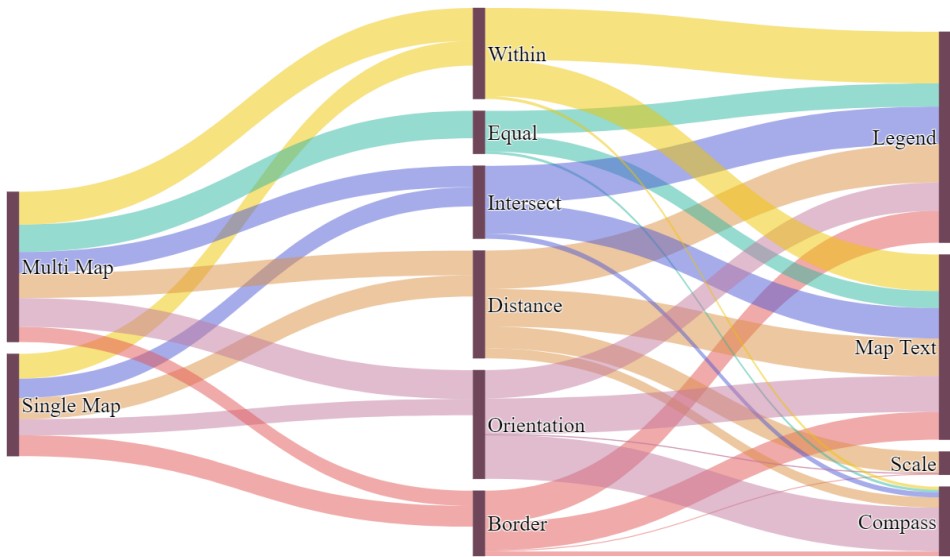

Figure 11: Sankey diagram illustrating the hierarchical structure of FRIEDA's question taxonomy. Each question is annotated with its count of maps (left), the spatial relation that defines the core reasoning objective (middle), and the specific map elements that must be interpreted to answer the question (right). The flow demonstrates how the dimensions interact in practice, highlighting that FRIEDA's questions typically require reasoning across multiple cartographic primitives.

| Spatial Relation | Map Count | Question Example |
|---|---|---|
| Border | Single | Which DMMUs along the Inner Harbor Navigation Canal share a boundary with 'DMMU 4'? Answer in the form DMMU #. |
| | Multi | Identify the 'National Road [map1]' that crosses the 'Ou Ta Paong' area. Which two 'Irrigation Schemes [map2]' does this road serve as a border for? Provide the names without the word 'Area'. |
| Distance | Single | What is the approximate straight-line distance between the SLC-6 Launch Site and the 2 psf contour of the Falcon Heavy Launch line? |
| | Multi | In Tinian, each 'Heritiera longipetiolata' species observation area is associated with a name. Which 'observation area[map1]' is located closest to the Noise Sensitive Receptor labeled 'T15[map2]'? |
| Equal | Multi | Which 'feature[map1]' on the infrastructure map corresponds to 'Existing Component 60[map2]' of the Santander project? |
| Intersect | Single | How many Asanko tenement blocks does the Haul Road intersect? |
| | Multi | Which 'claim block(s)[map1]' of the UEX Christie Lake Project are crossed by the 'power line[map2]'? |
| Orientation | Single | What is the name of the northernmost 'National Air Monitoring Site' as recorded by Ordnance Survey Ireland? |
| | Multi | In the Lumberton Loop Project Area, what is the orientation of the 'Crosswalk Stripping [map1]' in relation to the 'Walnut Street Component[map2]'? |
| Within | Single | Along Pine Street and Pike Street, how many 'Future Redevelopment & Renovation Project' areas overlap with the 'West Focus Area'? |
| | Multi | Identify the area of Nighthawk Gold Property located North of the 'Winter Road[map1]'. How many 'Gold Deposits[map2]' are located within this area? |

Table 5: Example questions by spatial relation and map count. For multi-map questions, entities are annotated with [map1]/[map2] only for illustration, indicating different source maps; these tags are not part of the actual questions. We underline the word/phrase that denotes the target spatial relation.

**FRIEDA Coverage: 32 Countries**

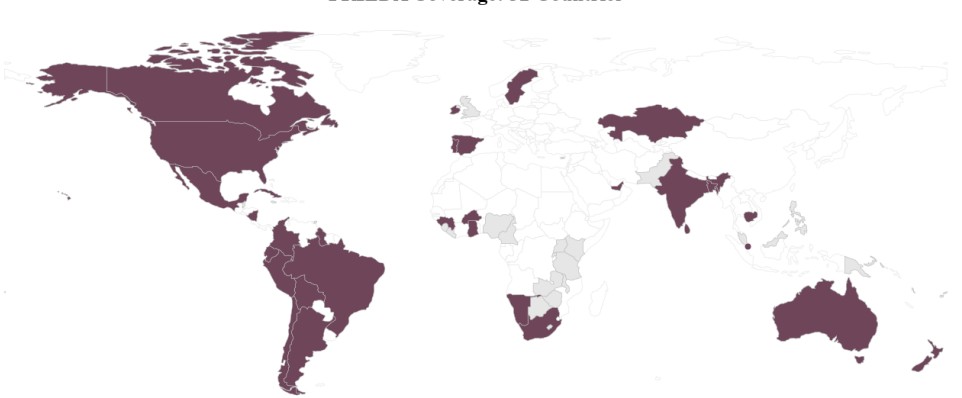

Figure 12: Global country coverage of FRIEDA. Countries included in the dataset are shown in purple; countries where English is a primary or official working language but not covered by FRIEDA are shaded light gray. Coverage spans six continents (32 countries).

**Domain Coverage**    We source maps from domains where spatial reasoning is essential: geologic reports (SEDAR+, 2025), national park management reports (National Park Service, 2025), investment and infrastructure reports (AIIB, 2025), disaster and hazard assessments (Federal Emergency Management Agency, 2025), city and regional planning documents (Seattle Planning and Commu-

| Country | Count | Country | Count |
|---|---|---|---|
| United States | 251 | Mexico | 18 |
| Canada | 82 | Portugal | 2 |
| South Africa | 32 | New Zealand | 1 |
| Peru | 9 | Chile | 4 |
| Burkina Faso | 1 | Brazil | 2 |
| Guyana | 2 | Guinea | 3 |
| Ireland | 24 | Colombia | 2 |
| Seychelles | 14 | Ecuador | 1 |
| Singapore | 9 | Cuba | 1 |
| Kazakhstan | 6 | Argentina | 3 |
| Cambodia | 5 | Bolivia | 2 |
| India | 7 | Spain | 1 |
| Bangladesh | 6 | Sweden | 1 |
| Sri Lanka | 3 | Australia | 1 |
| United Arab Emirates | 3 | Namibia | 2 |
| Ghana | 1 | Nicaragua | 1 |

Table 6: Country coverage in FRIEDA. Count reflects the number of questions whose maps originate from each country.

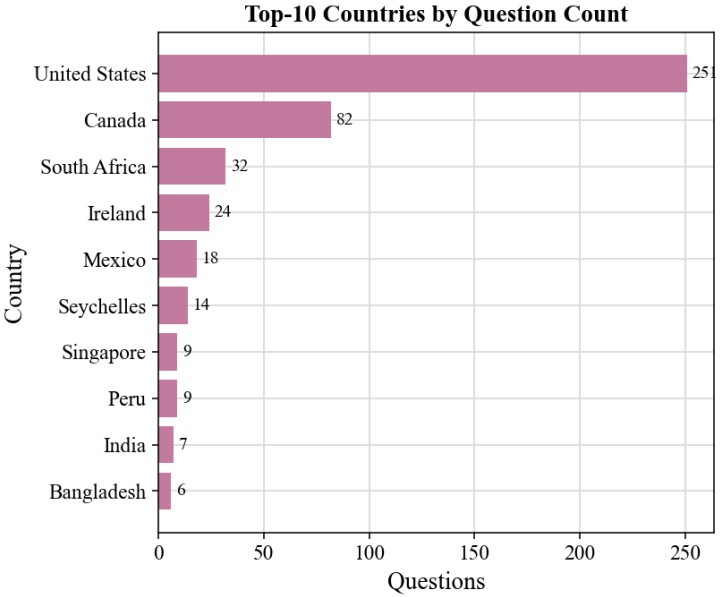

Figure 13: Top 10 countries by question Count

nity Development, 2024; City of Cape Town, 2025; Department of Community Development, 2025; Urban Redevelopment Authority, 2025), and environmental reviews (U.S. Environmental Protection Agency, 2025; Ministry of Agriculture Climate Change and Environment, 2025; Environmental Protection Agency, 2025). Several of these are umbrella categories that can be further subdivided. For example, environmental assessments may target facilities, hydrology, land use/land cover, or habitat. For consistency, we retain the top-level labels used by the source repositories. Across these domains, maps employ varied symbol conventions (legends, scale bars, north arrows) and heterogeneous geometry types (areas, lines, points), encouraging generalization beyond any single map style. Figure 14 summarizes the domain coverage.

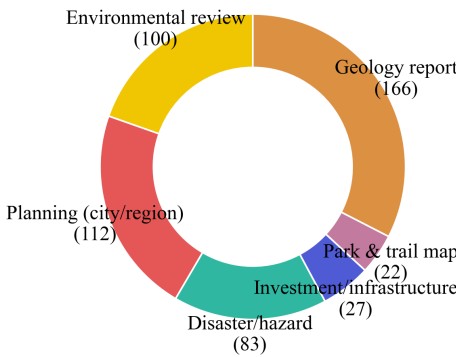

Figure 14: Domain distribution in FRIEDA by document category (e.g., geology, planning). Slices indicate categories, and parentheses denote question counts.

## D    EXAMPLES OF FRIEDA

We store each example as a JSON record containing the question, the gold answer, the required map image(s), any contextual image(s), and metadata such as the number of maps, target spatial relation, and answer type. Figure 15 illustrates an example of a single-map question, and Figure 16 shows an example of a multi-map question.

```
"question_ref":"q_1093",
"question_text":"What is the orientation of 'Bryan Palmer & Barry Maust' in relation to
'Gary Blocher' within the Meyersdale Study Area?",
"expected_answer":"South",
"image_urls":[
  "EIS/Vol-3-FEISAppendixA-M-May-2025/image21_m1.png"
],
"map_count":"Single",
"spatial_relationship":"Orientation",
"answer_type":"cardinal",
"contextual_urls":[
  "EIS/Vol-3-FEISAppendixA-M-May-2025/image21_m1.png",
  "EIS/Vol-3-FEISAppendixA-M-May-2025/image21_m0.png",
  "EIS/Vol-3-FEISAppendixA-M-May-2025/image20_1.png",
  "EIS/Vol-3-FEISAppendixA-M-May-2025/image22_1.png",
  "EIS/Vol-3-FEISAppendixA-M-May-2025/image19_1.png",
  "EIS/Vol-3-FEISAppendixA-M-May-2025/image26_1.png",
  "EIS/Vol-3-FEISAppendixA-M-May-2025/image15_1.png",
  "EIS/Vol-3-FEISAppendixA-M-May-2025/image11_1.png",
  "EIS/Vol-3-FEISAppendixA-M-May-2025/image10_1.png",
  "EIS/Vol-3-FEISAppendixA-M-May-2025/image12_1.png"
],
"domain":"Environmental review",
"map_elements":[
  "map text",
  "compass"
]
```

Figure 15: Example question single map

## E    DETAILED BENCHMARK RESULT AND ANALYSIS

### E.1    LLM-AS-JUDGE PROMPT

To evaluate free-form textual answers, we employ LLM-as-Judge (Gu et al., 2025) using Mistral-Small-3.1 (Mistral AI, 2024). Since not all models follow our requested output format ("Final answer: ⟨your answer⟩") and minor wording differences may occur (e.g., '15.00%' vs. '15'), we

```
"question_ref":"q_0150",
"question_text":"The Aberdeen-Hoquiam North Shore Levee is classified into three
categories.  In which category is the 'Hoquiam Police Station' located?",
"expected_answer":"North Shore Levee (West)",
"image_urls":[
  "FEMA/BRIC-EMS-2020-BR-102-0002_WA-NorthShoreLeveeWest-DEA-20241126/image116_1.png",
  "FEMA/BRIC-EMS-2020-BR-102-0002_WA-NorthShoreLeveeWest-DEA-20241126/image101_1.png"
],
"map_count":"Multi",
"spatial_relationship":"Intersect",
"answer_type":"textual",
"contextual_urls":[
  "FEMA/BRIC-EMS-2020-BR-102-0002_WA-NorthShoreLeveeWest-DEA-20241126/image116_1.png",
  "FEMA/BRIC-EMS-2020-BR-102-0002_WA-NorthShoreLeveeWest-DEA-20241126/image118_1.png",
  "FEMA/BRIC-EMS-2020-BR-102-0002_WA-NorthShoreLeveeWest-DEA-20241126/image136_1.png",
  "FEMA/BRIC-EMS-2020-BR-102-0002_WA-NorthShoreLeveeWest-DEA-20241126/image138_1.png",
  "FEMA/BRIC-EMS-2020-BR-102-0002_WA-NorthShoreLeveeWest-DEA-20241126/image101_1.png",
  "FEMA/BRIC-EMS-2020-BR-102-0002_WA-NorthShoreLeveeWest-DEA-20241126/image139_1.png",
  "FEMA/BRIC-EMS-2020-BR-102-0002_WA-NorthShoreLeveeWest-DEA-20241126/image140_1.png",
  "FEMA/BRIC-EMS-2020-BR-102-0002_WA-NorthShoreLeveeWest-DEA-20241126/image141_1.png",
  "FEMA/BRIC-EMS-2020-BR-102-0002_WA-NorthShoreLeveeWest-DEA-20241126/image142_1.png",
  "FEMA/BRIC-EMS-2020-BR-102-0002_WA-NorthShoreLeveeWest-DEA-20241126/image137_1.png"
],
"domain":"Disaster/hazard",
"map_elements":[
  "legend"
]
```

Figure 16: Example question multi map

first require the LLM to extract the answer span based on the question and then compare the extracted portion to the gold answer with tolerance for minor variants (Figure 17).

> You will be given a triple consisting of a question, an expected answer, and a given response. Your task is to output either 'yes' or 'no'. Given the question and response, extract only the exact portion of the text that serves as the answer from the given response. Then output 'yes' if the user response conveys the same meaning as the expected answer in relation to the question. Output 'no' if it does not. For questions with multiple correct answers, the expected answers are separated by semicolons. The user response is correct if it matches all required answers, regardless of order. When the user provides more items than required, the response is treated as incorrect. If the user lists fewer items than expected, mark the response as incorrect. Differences in plurality, extra details such as acronyms or counts, minor typographical errors, and differences in wording style do not affect correctness. Focus only on whether the meaning matches.
>
> Question: {Question}
> Expected answer: {Expected Answer}
> Given response: {User Response}
>
> Does the response correctly answer the question based on the expected answer?
> Answer strictly 'yes' or 'no'

Figure 17: The input prompt to generate questions.

### E.2 STATISTICAL SIGNIFICANCE OF FRIEDA-DIRECT RESULTS

As FRIEDA partitions questions into a large number of fine-grained categories, some subsets contain relatively few examples (fewer than 100). In such cases, raw accuracy comparisons can be unreliable due to limited sample size. To more rigorously assess whether observed performance differences within these smaller subcategories are statistically meaningful, we apply McNemar's test McNEMAR (1947) on the top three proprietary models. We use the exact binomial version of the test when the number of disagreements is small ($< 50$), and the $\chi$-squared version with correction when disagreements are larger ($\geq 50$). Table 7 reports the resulting $p$-values.

| Category | Gemini 2.5 vs. GPT-5 | Gemini 2.5 vs. Sonnet-4 | GPT-5 vs. Sonnet-4 |
|---|---|---|---|
| Single-Map | **0.03** | **0.01** | 1.00[*] |
| Multi-Map | **0.05**[*] | **0.02** | **<0.01** |
| Border | 0.17 | 1.00[*] | 0.10[*] |
| Distance | 0.80[*] | 0.84 | 0.54 |
| Equal | 0.10[*] | 0.63[*] | 0.21 |
| Intersect | 0.50[*] | 0.13 | 0.03 |
| Orientation | 0.82 | **0.02** | **0.08** |
| within | 0.11 | **<0.01** | 0.09 |

Table 7: $p$-values from pairwise McNemar's tests across key subcategories. Bold values indicate statistical significance at $\alpha = 0.05$. An asterisk ([*]) indicates that for each pair (A vs. B), model B achieved a higher accuracy.

### E.3 PERFORMANCE ON FRIEDA-CONTEXTUAL

Table 8 reports overall and per-spatial relation performance for FRIEDA-contextual. As noted in Section 5, models show little difference between the FRIEDA-direct and FRIEDA-contextual settings. Figure 18 summarizes overall accuracy across models on FRIEDA-contextual.

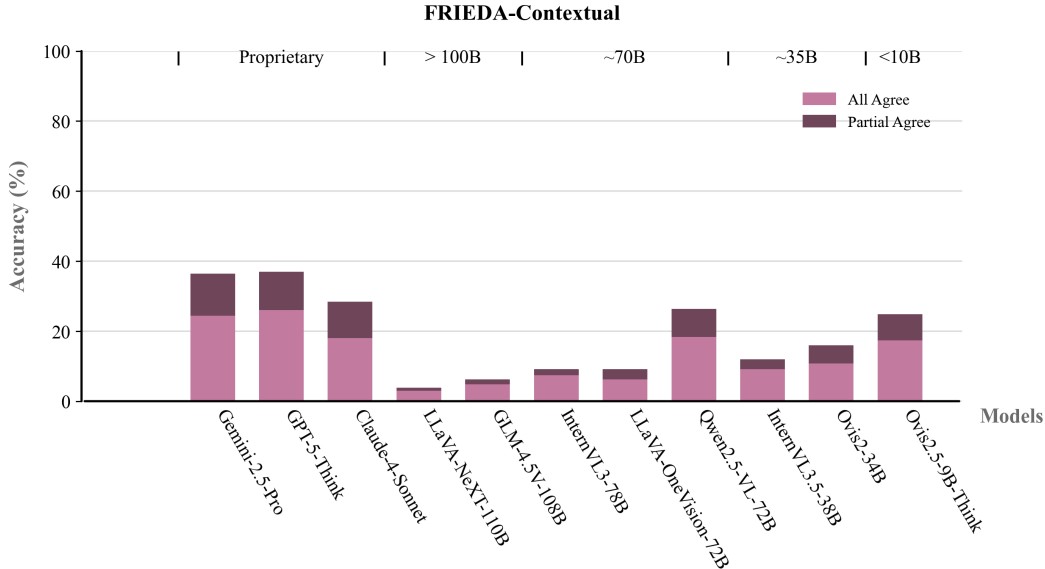

Figure 18: Overall accuracy across models in the FRIEDA-contextual setting.

### E.4 PERFORMANCE ON ALL-AGREE SUBSET

To validate that the performance gap reported in Section 5 is not an artifact of annotation noise, we evaluate models not only on the full dataset but also on the All-agree subset, where all three annotators unanimously agreed on the gold answer. Table 9 presents the results for both the full dataset and the All-agree subset, for both the direct and contextual setting of FRIEDA.

### E.5 PER MAP COUNT & ANSWER TYPE RESULT BREAKDOWN

We also report performance by map count and answer type for FRIEDA-direct (Table 10) and FRIEDA-contextual (Table 11). In the FRIEDA-direct setting, GPT-5-Think leads on multi-map questions, outperforming the next-best model (Gemini-2.5-Pro) by roughly 5%. Claude-Sonnet-4 performs best on *Distance* answers but underperforms on directional (i.e., *Orientation*) questions.

| | Overall (500) | Border (71) | Distance (91) | Equal (54) | Intersect (80) | Orientation (89) | Within (115) |
|---|---|---|---|---|---|---|---|
| | *Proprietary LVLMs* | | | | | | |
| Gemini 2.5 Pro | 36.60 | **28.17** | **29.67** | **50.00** | 21.25 | **64.77** | **30.17** |
| GPT-5-Think | **37.00** | **28.17** | 27.47 | 40.74 | **36.25** | 61.36 | **30.17** |
| Claude Sonnet 4 | 28.40 | 19.72 | 19.78 | 27.78 | 23.75 | 55.68 | 23.28 |
| | *Open Source LVLMs* | | | | | | |
| LLaVA-NeXT-110B | 3.80 | 2.86 | 3.30 | 3.64 | 7.50 | 0.00 | 5.17 |
| GLM-4.5V-108B | 7.40 | 9.46 | 0.00 | 6.00 | 13.41 | 1.12 | 12.82 |
| InternVL3-78B | 9.20 | 2.82 | 5.49 | 5.56 | 3.75 | 30.68 | 5.17 |
| LLaVA-OneVision-72B | 9.20 | 7.04 | 5.49 | 3.7 | 7.5 | 17.05 | 11.21 |
| Qwen2.5-VL-72B | 26.40 | 12.68 | 16.48 | 29.63 | 16.25 | 55.68 | 25.86 |
| InternVL3-78B | 9.20 | 2.82 | 5.49 | 5.56 | 3.75 | 30.68 | 5.17 |
| InternVL3.5-38B | 12.00 | 8.45 | 4.40 | 7.41 | 7.50 | 34.09 | 8.62 |
| Ovis2-34B | 16.00 | 21.13 | 14.29 | 18.52 | 18.75 | 2.27 | 21.55 |
| Ovis2.5-9B-Think | 24.80 | 18.31 | 9.89 | 27.78 | 21.25 | 56.82 | 17.24 |

Table 8: Performance of the 11 LVLMs across 6 spatial relationships on FRIEDA-contextual setting. Values represent performance scores (in %) for each spatial relationship and the overall accuracy.

| | FRIEDA-direct | | FRIEDA-contextual | |
|---|---|---|---|---|
| | Full (500) | All-agree (297) | Full (500) | All-agree (297) |
| Human Average | **84.87** | **93.93**[*] | - | - |
| | *Proprietary LVLMs* | | | |
| Gemini 2.5 Pro | **38.20** | **46.13** | **33.06** | 15.56 |
| GPT-5-Think | 37.20 | 44.11 | 30.65 | 26.67 |
| Claude Sonnet 4 | 31.60 | 24.26 | 25.81 | **28.89** |
| | *Open Source LVLMs* | | | |
| LLaVA-NeXT-110B | 8.60 | 9.43 | 10.48 | 8.89 |
| GLM-4.5V-108B | 6.40 | 8.67 | 8.33 | 0.00 |
| InternVL3-78B | 11.00 | 13.80 | 6.18 | 4.44 |
| LLaVA-OneVision-72B | 13.00 | 14.48 | 9.41 | 11.11 |
| Qwen2.5-VL-72B | 25.60 | 28.28 | 21.24 | 8.89 |
| InternVL3.5-38B | 14.20 | 14.81 | 9.68 | 6.67 |
| Ovis2-34B | 17.80 | 20.54 | 22.58 | 11.11 |
| Ovis2.5-9B-Think | 25.80 | 29.97 | 20.43 | 20.00 |

[*]Note: Although the All-agree subset reflects complete human consensus on the ground truth, the human average score is 93.33% rather than 100% because our evaluation pipeline relies on an LLM-as-Judge. In other words, the 93.33% accuracy reflects the LLM Judge's assessment of the human-provided answer on the All-agree items, not human disagreement.

Table 9: Performance of humans and 11 LVLMs on the All-agree subset for FRIEDA-direct and FRIEDA-contextual.

| | Map Count | | | Answer Types | | |
|---|---|---|---|---|---|---|
| | Overall (500) | Single (202) | Multi (298) | Textual (372) | Distance (45) | Direction (83) |
| Human Average | **84.87** | **84.91** | **88.08** | **87.93** | **67.18** | **92.15** |
| *Proprietary LVLMs* | | | | | | |
| Gemini 2.5 Pro | **38.20** | **32.67** | 41.95 | **33.06** | 15.56 | **73.49** |
| GPT-5-Think | 37.20 | 23.76 | **46.31** | 30.65 | 26.67 | 72.29 |
| Claude Sonnet 4 | 31.60 | 24.26 | 36.58 | 25.81 | **28.89** | 59.04 |
| *Open Source LVLMs* | | | | | | |
| LLaVA-NeXT-110B | 8.60 | 7.43 | 9.40 | 10.48 | 8.89 | 0.00 |
| GLM-4.5V-108B | 6.40 | 4.81 | 7.53 | 8.33 | 0.00 | 1.23 |
| InternVL3-78B | 11.00 | 6.93 | 13.76 | 6.18 | 4.44 | 36.14 |
| LLaVA-OneVision-72B | 13.00 | 15.35 | 11.41 | 9.41 | 11.11 | 30.12 |
| Qwen2.5-VL-72B | 25.60 | 21.78 | 28.19 | 21.24 | 8.89 | 54.22 |
| InternVL3.5-38B | 14.20 | 11.88 | 15.77 | 9.68 | 6.67 | 38.55 |
| Ovis2-34B | 17.80 | 17.33 | 18.12 | 22.58 | 11.11 | 0.00 |
| Ovis2.5-9B-Think | 25.80 | 22.28 | 28.19 | 20.43 | 20.00 | 53.01 |

Table 10: Performance of humans and 11 LVLMs across the two map count types and three answer types on FRIEDA-direct.

| | Map Count | | | Answer Types | | |
|---|---|---|---|---|---|---|
| | Overall (500) | Single (202) | Multi (298) | Textual (372) | Distance (45) | Direction (83) |
| *Proprietary LVLMs* | | | | | | |
| Gemini 2.5 Pro | 36.60 | 25.25 | **44.30** | **31.99** | 17.78 | **67.47** |
| GPT-5-Think | **37.00** | **26.24** | **44.30** | 31.72 | **28.89** | 65.06 |
| Claude Sonnet 4 | 28.40 | 20.30 | 33.89 | 23.12 | 17.78 | 57.83 |
| *Open Source LVLMs* | | | | | | |
| LLaVA-NeXT-110B | 3.80 | 1.99 | 5.03 | 4.85 | 2.22 | 0.00 |
| GLM-4.5V-108B | 7.40 | 6.19 | 8.28 | 9.95 | 0.00 | 0.00 |
| InternVL3-78B | 9.20 | 6.93 | 10.74 | 4.57 | 4.44 | 32.53 |
| LLaVA-OneVision-72B | 9.20 | 7.43 | 10.40 | 7.53 | 6.67 | 18.07 |
| Qwen2.5-VL-72B | 26.40 | 18.32 | 31.88 | 21.24 | 11.11 | 57.83 |
| InternVL3.5-38B | 12.00 | 8.42 | 14.43 | 7.53 | 4.44 | 36.14 |
| Ovis2-34B | 16.00 | 14.36 | 17.11 | 19.62 | 15.56 | 0.00 |
| Ovis2.5-9B-Think | 24.80 | 20.79 | 27.52 | 19.62 | 6.67 | 57.83 |

Table 11: Performance of the 11 LVLMs across the two map count types and three answer types on FRIEDA-contextual.

### E.6 PER MAP ELEMENT & MAP ELEMENT COUNT RESULT BREAKDOWN

We analyze performance based on the specific map elements required to answer each question, as well as the number of distinct element types involved, for both FRIEDA-direct (Table 12) and FRIEDA-contextual (Table 13). As map elements are not mutually exclusive, a single question may require interpreting multiple elements simultaneously to produce a correct answer.

Humans outperform every model by a large margin across all four map elements. Accuracy is highest when only one or two elements are required, but drops substantially when four elements must be combined, indicating that even expert map readers experience increased difficulty as compositional complexity grows. On the other hand, the best proprietary model performance occurs at three elements; this may be because questions involving multiple components compel the model to search the image to identify relevant elements.

| | Map Element Type | | | | Map Element Count | | | |
|---|---|---|---|---|---|---|---|---|
| | Map text (366) | Legend (417) | Compass (137) | Scale (46) | 1 (132) | 2 (279) | 3 (80) | 4 (9) |
| Human Average | **80.97** | **83.61** | **75.91** | **63.78** | **84.09** | **81.84** | **80.00** | 51.85 |
| *Proprietary LVLMs* | | | | | | | | |
| Gemini 2.5 Pro | **38.80** | **37.41** | **56.20** | 17.39 | **35.61** | **35.13** | 55.00 | 22.22 |
| GPT-5-Think | 38.52 | **34.05** | 53.28 | 28.26 | **36.36** | 34.41 | 48.75 | 33.33 |
| Claude Sonnet 4 | 31.69 | 31.41 | 51.83 | **30.43** | 24.24 | 29.75 | 47.50 | **55.56** |
| *Open Source LVLMs* | | | | | | | | |
| LLaVA-NeXT-110B | 7.38 | 8.87 | 0.73 | 8.70 | 14.39 | 8.24 | 0.00 | 11.11 |
| GLM-4.5V-108B | 5.72 | 5.74 | 3.62 | 0.00 | 12.12 | 5.00 | 2.47 | 0.00 |
| InternVL3-78B | 9.84 | 10.31 | 23.36 | 4.35 | 9.85 | 9.32 | 20.00 | 0.00 |
| LLaVA-OneVision-72B | 13.39 | 11.27 | 20.44 | 10.87 | 10.61 | 13.62 | 16.25 | 0.00 |
| Qwen2.5-VL-72B | 26.23 | 24.46 | 40.88 | 10.87 | 21.97 | 24.37 | 37.50 | 11.11 |
| InternVL3.5-38B | 14.48 | 12.23 | 25.55 | 6.52 | 12.12 | 14.34 | 17.50 | 11.11 |
| Ovis2-34B | 16.12 | 18.47 | 4.38 | 10.87 | 27.27 | 17.20 | 6.25 | 0.00 |
| Ovis2.5-9B-Think | 26.78 | 23.26 | 38.69 | 21.74 | 23.48 | 24.73 | 33.75 | 22.22 |

Table 12: Performance of humans and 11 LVLMs across the map element types and count of map elements on FRIEDA-direct.

| | Map Element Type | | | | Map Element Count | | | |
|---|---|---|---|---|---|---|---|---|
| | Map text (366) | Legend (417) | Compass (137) | Scale (46) | 1 (132) | 2 (279) | 3 (80) | 4 (9) |
| *Proprietary LVLMs* | | | | | | | | |
| Gemini 2.5 Pro | 37.43 | **35.49** | 50.36 | 19.57 | **34.85** | 34.77 | 46.25 | **33.33** |
| GPT-5-Think | **38.25** | 34.53 | **51.09** | **30.43** | 34.09 | **35.48** | 48.75 | 22.22 |
| Claude Sonnet 4 | 29.78 | 26.38 | 43.07 | 19.57 | 25.00 | 27.24 | 37.50 | **33.33** |
| *Open Source LVLMs* | | | | | | | | |
| LLaVA-NeXT-110B | 3.01 | 4.32 | 0.00 | 2.17 | 6.06 | 3.94 | 0.00 | 0.00 |
| GLM-4.5V-108B | 5.93 | 6.75 | 4.67 | 0.00 | 10.95 | 6.13 | 3.33 | 0.00 |
| InternVL3-78B | 8.47 | 8.39 | 20.44 | 4.35 | 6.06 | 9.32 | 15.00 | 0.00 |
| LLaVA-OneVision-72B | 7.65 | 9.11 | 13.87 | 6.52 | 9.85 | 8.96 | 8.75 | 11.11 |
| Qwen2.5-VL-72B | 27.32 | 24.46 | 41.61 | 13.04 | 21.97 | 26.88 | 32.50 | 22.22 |
| InternVL3.5-38B | 13.39 | 10.31 | 23.36 | 4.35 | 6.06 | 13.62 | 17.50 | 0.00 |
| Ovis2-34B | 13.93 | 16.31 | 5.11 | 15.22 | 25.00 | 14.70 | 7.50 | 0.00 |
| Ovis2.5-9B-Think | 26.23 | 22.30 | 40.15 | 8.70 | 21.97 | 24.01 | 33.75 | 11.11 |

Table 13: Performance of the 11 LVLMs across the map element types and count of map elements on FRIEDA-contextual.

### E.7    PER DOMAIN RESULT BREAKDOWN

In addition, we report performance by domain for FRIEDA-direct (Table 14) and FRIEDA-contextual (Table 15). The domain can serve as an indicator of map style heterogeneity. For example, reports from park maps (labeled "Parks" in the table) and disaster reports typically follow the same formalized format because they are produced by the same source (usually the government). In contrast, reports from the investment and infrastructure domain (labeled "Investment") and the geology domain (labeled "Geology") originate from various sources, as they are usually authored by different companies, resulting in more diverse map styles.

| | Overall (500) | Planning (112) | Investment (27) | Environment (100) | Disaster (83) | Parks (22) | Geology (166) |
|---|---|---|---|---|---|---|---|
| Human Average | **84.87** | **86.60** | **88.89** | **82.33** | **83.13** | **75.76** | **76.91** |
| *Proprietary LVLMs* | | | | | | | |
| Gemini 2.5 Pro | **38.20** | **37.25** | **33.33** | **43.00** | 49.40 | 45.45 | **30.12** |
| GPT-5-Think | 37.20 | 35.29 | 25.93 | 40.00 | **54.22** | **68.18** | 25.90 |
| Claude Sonnet 4 | 31.60 | 33.33 | 22.22 | 28.00 | 42.17 | 50.00 | 26.51 |
| *Open Source LVLMs* | | | | | | | |
| LLaVA-NeXT-110B | 8.60 | 9.80 | 18.52 | 6.00 | 8.43 | 13.64 | 7.23 |
| GLM-4.5V-108B | 6.40 | 3.92 | 0.00 | 5.00 | 8.43 | 9.09 | 8.33 |
| InternVL3-78B | 11.00 | 12.75 | 7.41 | 12.00 | 16.87 | 22.73 | 5.42 |
| LLaVA-OneVision-72B | 13.00 | 16.67 | 11.11 | 11.00 | 7.23 | 22.73 | 13.86 |
| Qwen2.5-VL-72B | 25.60 | 29.41 | 18.52 | 21.00 | 34.94 | 22.73 | 22.89 |
| InternVL3.5-38B | 14.20 | 13.73 | 22.22 | 15.00 | 18.07 | 18.18 | 10.24 |
| Ovis2-34B | 17.80 | 18.63 | 14.81 | 19.00 | 21.69 | 22.73 | 14.46 |
| Ovis2.5-9B-Think | 25.80 | 21.57 | 22.22 | 23.00 | 33.73 | 40.91 | 24.70 |

Table 14: Performance of humans and 11 LVLMs across the seven domain types on FRIEDA-direct.

| | Overall (500) | Planning (112) | Investment (27) | Environment (100) | Disaster (83) | Parks (22) | Geology (166) |
|---|---|---|---|---|---|---|---|
| *Proprietary LVLMs* | | | | | | | |
| Gemini 2.5 Pro | 36.60 | **39.22** | **40.74** | **34.00** | **50.60** | 50.00 | 27.11 |
| GPT-5-Think | **37.00** | 36.27 | 25.93 | **34.00** | 49.40 | **72.73** | **30.12** |
| Claude Sonnet 4 | 28.40 | 30.39 | 25.93 | 24.00 | 42.17 | 45.45 | 21.08 |
| *Open Source LVLMs* | | | | | | | |
| LLaVA-NeXT-110B | 3.80 | 3.92 | 14.81 | 3.00 | 7.23 | 4.55 | 0.60 |
| GLM-4.5V-108B | 7.40 | 5.61 | 0.00 | 5.66 | 10.23 | 0.00 | 8.29 |
| InternVL3-78B | 9.20 | 8.82 | 7.41 | 12.00 | 13.25 | 18.18 | 4.82 |
| LLaVA-OneVision-72B | 9.20 | 12.75 | 14.81 | 9.00 | 10.84 | 18.18 | 4.22 |
| Qwen2.5-VL-72B | 26.40 | 25.49 | 37.04 | 20.00 | 44.58 | 36.36 | 18.67 |
| InternVL3.5-38B | 12.00 | 10.78 | 14.81 | 13.00 | 15.66 | 27.27 | 7.83 |
| Ovis2-34B | 16.00 | 16.67 | 25.93 | 12.00 | 22.89 | 18.18 | 12.65 |
| Ovis2.5-9B-Think | 24.80 | 20.59 | 18.52 | 25.00 | 36.14 | 36.36 | 21.08 |

Table 15: Performance of the 11 LVLMs across the seven domain types on FRIEDA-contextual.

## F    EXTENDED ANALYSES

### F.1    EXAMPLES OF EACH ERROR CATEGORY

We illustrate the three most frequent error categories for Gemini-2.5-Pro and show each example alongside answers and reasoning from Gemini-2.5-Pro, GPT-5-Think, and Claude-Sonnet-4.

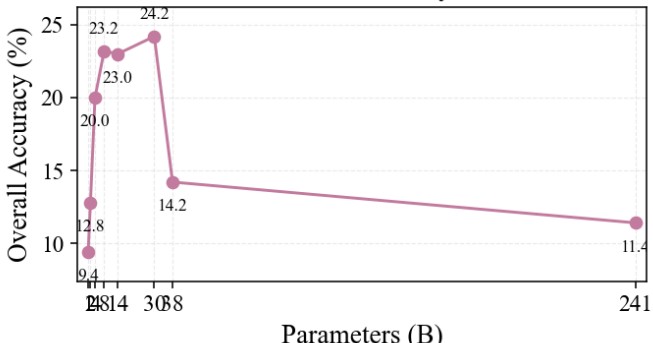

| Model Size | Accuracy (%) |
|------------|--------------|
| 1B | 9.40 |
| 2B | 12.80 |
| 4B | 20.00 |
| 8B | 23.20 |
| 14B | 23.00 |
| **30BA3B** | **24.20** |
| 38B | 14.20 |
| 241BA28B | 11.40 |

Table 16: InternVL3.5 performance by size

Figure 19: Performance of InternVL3.5 by model parameter size

**Misinterpretation of legend**  Listing 1 presents a case where the model fails to map a legend symbol or color to its intended semantic class, leading to the selection of the wrong feature despite the correct evidence being present.

**Cross-map interpretation failure**  Listing 2 shows a failure that arises when reasoning requires aligning information across multiple maps or overlays; the model identifies the wrong subject when the maps must be cross-referenced.

**Spatial-relation semantics error**  Listing 3 illustrates a case where the model misinterprets the key spatial relation, yielding an incorrect answer.

## F.2    ANALYSES ON MODEL SIZE

In the main evaluation (Figure 3), the results deviate from the usual scaling law (Kaplan et al., 2020), which states that the performance of the model improves with size. Among open-source models, LLaVA-NeXT, despite having the most parameters, ranks near the bottom, whereas Ovis-2.5-9B, the smallest model, ranks near the top. We, therefore, hypothesize that cartographic reasoning is not an emergent ability (i.e., a capability absent in smaller models but present in larger ones). To test this, we evaluate the InternVL3.5 family (Wang et al., 2025b) on FRIEDA: 1B, 2B, 4B, 8B, 14B, 30BA3B, 38B, 241BA28B where 'A' denotes parameters active at inference. The trend (Figure 19, Table 16) shows modest gains up to roughly 30B parameters, followed by degradation thereafter.

## G    EXTENDED RELATED WORKS

To provide a comprehensive context for FRIEDA, we detail the scope of related benchmarks across three areas: general document and infographics understanding, map visual question answering, and broader spatial reasoning.

### G.1    DOCUMENT AND INFOGRAPHICS VQA

Benchmarks in this domain have established baselines for LVLM reasoning over structured text and graphical images, including charts. In the document domain, DocVQA (Mathew et al., 2021) introduces a large-scale question-answering dataset over real forms and reports. DocVXQA (Souibgui et al., 2025) builds upon the benchmark to design a self-explanatory framework that produces interpretable rationales for LVLMs. Docopilot (Duan et al., 2025) evaluates LVLMs on scientific articles, which not only test text understanding but also the interpretation of embedded figures such as charts. For graphics, InfographicsVQA (Mathew et al., 2022) tests joint reasoning over text, layout, and pictorial elements in visually rich infographics. InfoChartQA (Lin et al., 2025) extends this by pairing plain charts and infographics and identifying design elements that degrade LVLM performance. In general, VQA evaluation on frontier LVLMs reveals a consistent trend: competence at

high-level patterns, such as identifying trends in the data or the extrema, but struggles with precise value extraction and robustness. FRIEDA evaluates these shortcomings in a cartographic setting, where layout, symbols, legends, scales, and compass orientation interact tightly to measure how well LVLMs integrate these signals to answer map-based questions.

## G.2 MAP VQA

Research in map understanding can be categorized into general map VQA, navigation-centered reasoning, and domain-specific question-answering.

**General Map VQA** MapQA (Chang et al., 2022) establishes a baseline for choropleth map understanding by creating question-answer pairs targeting value retrieval and region identification. However, the dataset is limited to a single map type (i.e., choropleth maps) and geographically restricted to the United States, thereby limiting the diversity of cartographic styles and toponyms. MapWise (Mukhopadhyay et al., 2025) broadens geographic coverage to the United States, India, and China and introduces diverse question templates for probing relative spatial relationships; yet, it still relies solely on choropleth maps and remains constrained to single-map reasoning, which limits its ability to model real-world cartographic complexity. MapIQ (Srivastava et al., 2025) further advances visualization literacy by introducing cartograms and proportional-symbol maps, which are commonly used in analytical tasks. While the expanded map diversity is valuable, MapIQ's maps are generated using map-coloring tools rather than drawn from heterogeneous, noisy real-world documents. In contrast, FRIEDA explicitly captures this real-world variability by sourcing map images directly from government and scientific reports.

**Navigation-centered Reasoning** Benchmarks centered on navigation often require more complex reasoning than simple semantic retrieval, yet they tend to remain domain-narrow. MapEval (Dihan et al., 2025) evaluates LVLMs' geospatial reasoning through multiple-choice travel-planning questions spanning 180 cities. Still, it relies on standard web basemaps (e.g., Google Maps) whose clean, uniform designs lack the layered, domain-specific symbology (e.g., variable legends, irregular projections, and customized north arrows) often found in professional cartography. ReasonMap (Feng et al., 2025) moves beyond basemaps by using high-resolution transit maps and designing navigation tasks that closely simulate real-world subway routing, though its scope is restricted to transit systems. MapBench (Xing et al., 2025) evaluates LVLMs' spatial reasoning and chain-of-thought inference by testing outdoor navigation performance on diverse map types, such as park and trail maps. Despite their contributions, all of these benchmarks remain focused on navigation-centric tasks. In contrast, our benchmark generalizes spatial reasoning across six distinct spatial relations that extend well beyond navigation, capturing the broader landscape of cartographic reasoning required in professional and scientific contexts.

**Domain- and Task-specific QA** Specialized benchmarks address domain-specific needs or particular visual modalities, but they tend to trade breadth for depth. PEACE (Huang et al., 2025) introduces a benchmark focused on geologic map understanding and develops a framework for answering domain-specific questions, such as identifying lithologic units, fault lines, and structural patterns. While the benchmark and the approach is highly effective for geology-specific evaluation, the scope is limited to a single scientific domain, and it lacks the thematic diversity required for broader cartographic reasoning. CartoMark (Zhou et al., 2024) provides a wide range of maps across various styles, but its core task centers on simple pattern recognition, such as scene classification and text annotation. These tasks primarily test perceptual recognition and, in many cases, do not require reasoning at all. ReMI (Kazemi et al., 2025) offers a framework for multi-image reasoning that evaluates how models integrate and compare information across visual inputs. However, ReMI operates on natural images and uses simple web-based maps. Therefore, it does not assess the specialized challenges of multi-map cartographic reasoning, such as aligning heterogeneous legends, reconciling differing spatial scales, and interpreting mismatched orientations across maps. These capabilities form the core of FRIEDA's multi-map setting, which reflects real-world analytical scenarios where experts must synthesize information from multiple, heterogeneous cartographic sources.

To situate FRIEDA within the broader landscape of MapVQA benchmarks, we provide a comparative summary of existing works in Table 17. The table evaluates each dataset along four key dimensions: (1) the types of spatial abilities evaluated, (2) diversity of map elements (measured through

country and domain coverage), (3) whether multi-map reasoning is supported, and (4) whether a contextual setting is included to emulate real-world map-use scenarios.

We use orange checkmarks (✓) to indicate partial or limited coverage within a category. For example, in the topological relation category, we treat questions such as "how many points lie along the route to location A?" partially covering topological relation as such tasks contain the notion of *intersect* while it does not examine the relation with the depth or rigor as required in FRIEDA. Overall, the comparison highlights that prior MapVQA benchmarks tend to emphasize narrow task settings, limited spatial relations, or constrained map styles, whereas FRIEDA is designed to provide comprehensive, cross-domain evaluation that reflects the complexity of real-world cartographic reasoning.

| | Spatial Relation | | | Heterogeneity | | | |
|---|---|---|---|---|---|---|---|
| | Topological | Metric | Directional | Country | Domain | Multi-Map | Contextual |
| MapQA (Chang et al., 2022) | ✗ | ✗ | ✗ | 1 | 1 | ✗ | ✗ |
| CartoMark (Zhou et al., 2024) | ✗ | ✗ | ✗ | 13 | 7 | ✗ | ✗ |
| MapWise (Mukhopadhyay et al., 2025) | ✓ | ✗ | ✓ | 3 | 3 | ✗ | ✗ |
| MapIQ (Srivastava et al., 2025) | ✓ | ✗ | ✗ | 1 | 6 | ✗ | ✗ |
| MapBench (Xing et al., 2025) | ✗ | ✗ | ✓ | UNK | 9 | ✗ | ✗ |
| MapEval (Dihan et al., 2025) | ✓ | ✓ | ✓ | 54 | 1 | ✗ | ✗ |
| ReMi (Kazemi et al., 2025) | ✗ | ✗ | ✓ | 100? | 1 | ✓ | ✗ |
| PEACE (Huang et al., 2025) | ✓ | ✓ | ✓ | 2 | 1 | ✗ | ✗ |
| ReasonMap (Feng et al., 2025) | ✗ | ✗ | ✗ | 13 | 1 | ✗ | ✗ |
| **FRIEDA** | ✓ | ✓ | ✓ | 32 | 6 | ✓ | ✓ |

Note: ReMi (Kazemi et al., 2025) reports counts by city, not by country; consequently, the corresponding country total is less than 100.

Table 17: A comparison of FRIEDA with prior map VQA benchmarks. FRIEDA covers a broader set of map-reading abilities and exhibits greater geographic and thematic diversity.

### G.3 SPATIAL REASONING

Spatial reasoning benchmarks have advanced model capabilities in perception and localization, though often outside the cartographic domain. Benchmarks such as SpatialVLM (Chen et al., 2024a) and SpatialRGPT (Cheng et al., 2024) focus on natural images, testing a model's ability to reason about 2D and 3D spatial relationships, relative positions, and object dimensions in photographic scenes. In the geospatial domain, GeoChain (Yerramilli et al., 2025) enhances tasks such as geolocation by inducing step-by-step geographic reasoning to link visual cues to geographic entities. However, these works primarily rely on natural scene understanding or semantic knowledge retrieval and do not engage with the abstract symbolic conventions unique to maps. FRIEDA closes this gap by evaluating multi-step cartographic reasoning, in which models must not only perceive space but also decode specific symbolic rules to infer topological, metric, and directional relations across heterogeneous real-world maps.

## H THE USE OF LARGE LANGUAGE MODELS

We acknowledge the use of large language models (LLMs) for benchmark question curation, revision, and polishing of this paper. The details of usage, the exact prompt used, and all related information are provided in the main paper or appendices. All questions created with the assistance of a large language model have been verified and modified by the authors. The paper's main contribution remains with the authors.

Listing 1: Legend misinterpretation example of Gemini-2.5-Pro on FRIEDA. Other models are shown for reference. `Orange` : Task Instruction. `Green` : Correct Answer. `Red` : Incorrect Answer.

`System:` :
Answer the questions based on the following criteria:
General:
    * If question can be answered, write answer in short answer box
    * If answer is a text from the map, copy it as it appears

Numerical Answers:
    * Include units as indicated on the map (Don't convert 1200m to 1.2km)
    * If both map frame and ruler scale is available, use the ruler scale
    * If question asks for an area, use {unit}^2
    * Use numerical values (e.g., 4 instead of four)

Directional Answers:
    * Use 8 cardinal directions only: North, North East, East, South East, South, South West,
    West, North West
    * Write 'North' or 'South' before 'East' or 'West'
    * Notice that the north arrow compass do not always point upward

Multi-Part Answers:
    * Separate with semicolon (;) (e.g., Zone A; Zone B)

    Give the final answer in 'Final answer: <your answer>'
    Do not use online search.

`Images:` :

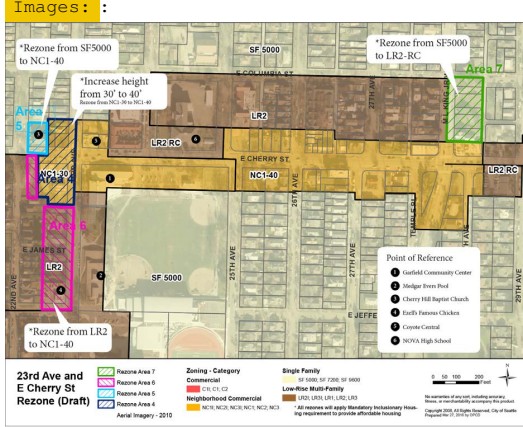

`Question:` The 23rd Ave and E Union St Rezone (Draft) classifies each area into four distinct Rezone Areas. With which Rezone Area does 'Rezone Area 4' share its westernmost boundary. Give the name of the area in form Rezone Area #.

`Gold Answer:` Rezone Area 6

`Gemini-2.5-Pro:`
I've determined the user's objective: to find the "Rezone Area" west of "Rezone Area 4". I've analyzed the map and deconstructed the task. "Rezone Area 4" is the blue-hatched area. Its western border is along 22nd Ave. I've identified the adjacent pink area. Now, according to the legend, the pink/magenta area corresponds to "Rezone Area 5". Final answer: Rezone Area 5

`GPT-5-Think:` The image shows a zoning map with areas labeled 4-7, and the title mentions "23 rd Ave and E Cherry St Rezone (Draft)." The question asks about Rezone Area 4's westernmost boundary. I'll need to find Area 4 on the map and check which area lies directly to the west of it, likely along 23rd Ave. The map shows colored boxes for Areas 4, 5, and 6. Area 4, labeled in light blue, is near the left margin, adjacent to "NC1-30" and "NC1-40." It seems that Area 4's westernmost boundary touches Area 5, which is directly to the left of Area 4. So , I suspect Area 5 shares this boundary. Final answer: Rezone Area 5

`Claude-4-Sonnet:` Looking at the map, I can identify Rezone Area 4 by its blue diagonal line pattern. This area is located in the western portion of the map.
Examining the westernmost boundary of Rezone Area 4, I can see that it directly borders the magenta/pink colored area, which according to the legend corresponds to Rezone Area 6. Final answer: Rezone Area 6

Listing 2: Cross-map interpretation failure example of Gemini-2.5-Pro on FRIEDA. Other models are shown for reference. Orange: Task Instruction. Green: Correct Answer. Red: Incorrect Answer.

System: :
```
Answer the questions based on the following criteria:
General:
    * If question can be answered, write answer in short answer box
    * If answer is a text from the map, copy it as it appears

Numerical Answers:
    * Include units as indicated on the map (Don't convert 1200m to 1.2km)
    * If both map frame and ruler scale is available, use the ruler scale
    * If question asks for an area, use {unit}^2
    * Use numerical values (e.g., 4 instead of four)

Directional Answers:
    * Use 8 cardinal directions only: North, North East, East, South East, South, South West,
    West, North West
    * Write 'North' or 'South' before 'East' or 'West'
    * Notice that the north arrow compass do not always point upward

Multi-Part Answers:
    * Separate with semicolon (;) (e.g., Zone A; Zone B)

    Give the final answer in 'Final answer: <your answer>'
    Do not use online search.
```

Images: :

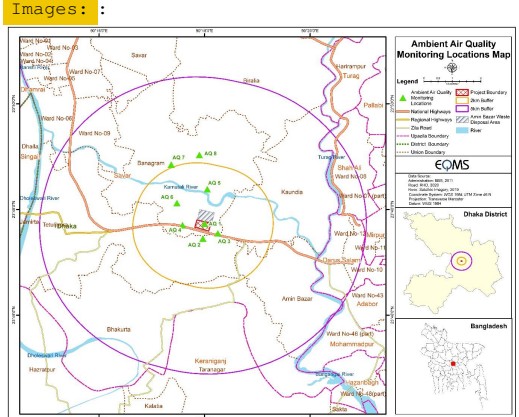 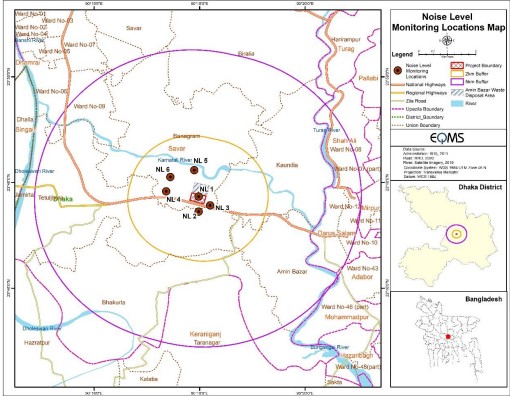

Question: Identify the Air Quality Monitoring Location closest to the Noise Level Monitoring Location 'NL 5'. What is the orientation of this point in relation to 'NL 5'?

Gold Answer: North East

———

Gemini-2.5-Pro: I've determined that AQ 6 is the closest point to NL 5. Now, I am analyzing their relative orientation. It appears that from NL 5, AQ 6 lies in a west-northwest direction. Final answer: AQ 6; West

GPT-5-Think: Looking at the maps, AQ5 seems nearest to NL5, based on their locations along the river. AQ5 seems to be slightly northeast of NL5, just a bit to the east of it. To answer the question, it makes sense to say AQ5 is closest to NL5. As for the orientation, I think AQ5 lies slightly southeast of NL5. The AQ5 triangle is located near the river, south of the river bend, while NL5 is to the north. I'll go with: AQ5 lies southeast of NL5. This seems like the most accurate conclusion. Final answer: AQ 5; southeast.

Claude-4-Sonnet: Looking at both maps, I need to locate NL 5 on the Noise Level Monitoring Locations Map and find the closest AQ location on the Ambient Air Quality Monitoring Locations Map. NL 5 is located near the Karnatali River area. Examining the AQ locations on the first map:
- AQ 5 is positioned very close to the same Karnatali River area
- AQ 6 is slightly further west
- AQ 2, AQ 3, AQ 4 are further south
- AQ 7 and AQ 8 are further north
AQ 5 is clearly the closest Ambient Air Quality Monitoring Location to NL 5.
Comparing the relative positions of AQ 5 and NL 5, AQ 5 appears to be positioned to the east of NL 5. Final answer: AQ 5; East"}

Listing 3: Spatial-relation semantics error example of Gemini-2.5-Pro on FRIEDA. Other models are shown for reference. `Orange`: Task Instruction. `Green`: Correct Answer. `Red`: Incorrect Answer.

```
System: :
Answer the questions based on the following criteria:
General:
    * If question can be answered, write answer in short answer box
    * If answer is a text from the map, copy it as it appears

Numerical Answers:
    * Include units as indicated on the map (Don't convert 1200m to 1.2km)
    * If both map frame and ruler scale is available, use the ruler scale
    * If question asks for an area, use {unit}^2
    * Use numerical values (e.g., 4 instead of four)

Directional Answers:
    * Use 8 cardinal directions only: North, North East, East, South East, South, South West,
    West, North West
    * Write 'North' or 'South' before 'East' or 'West'
    * Notice that the north arrow compass do not always point upward

Multi-Part Answers:
    * Separate with semicolon (;) (e.g., Zone A; Zone B)

    Give the final answer in 'Final answer: <your answer>'
    Do not use online search.
```

Images: :

Question: Which 'National Register Districts' overlap with both the 'Ballard' and 'Interbay Dravus' Industrial Lands Subareas within the Ballard-Interbay MIC?
Gold Answer: Washington Ship Canal

---

Gemini-2.5-Pro: I'm currently focused on dissecting the user's complex query. My goal is to determine the exact regions where the yellow "National Register Districts" overlap with both the purple "Ballard" and orange "Interbay Dravus" subareas, all contained within the "Ballard-Interbay MIC." I'm visualizing the overlay on the map. I've broken down the request into map elements and used the legend to visualize the features. I've concluded that the only district meeting all criteria is the "Chittenden Locks and Lake Washington Ship Canal." Final answer: Chittenden Locks and Lake Washington Ship Canal

GPT-5-Think:
I need to figure out which National Register Districts overlap with both the 'Ballard' and 'Interbay Dravus' subareas within the Ballard-Interbay MIC. I've got the map, and I think the overlapping yellow polygons (representing districts) fall within both the Ballard (pink?) and Interbay Dravus (green?) subareas. I'll check for yellow dots (properties) and orange polygons (districts), focusing on labels in the north area, like the Ballard Avenue Historic District. The "Washington Ship Canal" National Register District seems to be the most likely answer, as it follows the canal and overlaps both subareas. Final answer: Washington Ship Canal

Claude-4-Sonnet: Looking at the map, I need to identify National Register Districts (shown in orange) that overlap with both the Ballard Industrial Lands Subarea (shown in purple) and the Interbay Dravus Industrial Lands Subarea (shown in orange/peach). After careful examination of the map, I do not see any National Register Districts that overlap with both the Ballard and Interbay Dravus Industrial Lands Subareas within the Ballard-Interbay MIC.Final answer: None"

