# OpenReview forum: "FRIEDA: Benchmarking Multi-Step Cartographic Reasoning in Vision-Language Models"
_ICLR.cc/2026/Conference — ICLR 2026 Poster_

### Official Review · Reviewer_Y3WZ · 2025-10-30

**Soundness:** 3
**Presentation:** 4
**Contribution:** 3
**Rating:** 8
**Confidence:** 4

**Summary:**

This paper introduces FRIEDA, a new benchmark designed to evaluate multi-step cartographic reasoning in LVLMs. FRIEDA targets the full spectrum of spatial reasoning, topological (border, equal, intersect, within), metric (distance), and directional (orientation), using real-world maps drawn from public documents such as geological surveys, planning reports, and environmental studies. The benchmark includes 500 questions and 17,030 map images, testing models under two settings: a direct mode, where relevant maps are provided, and a contextual mode, where the model must identify the correct maps before reasoning. Results across 11 state-of-the-art LVLMs show that even the strongest systems (Gemini-2.5-Pro and GPT-5-Think) achieve less than 40% accuracy, far below human performance. The study highlights persistent challenges in spatial and multi-map reasoning, offering FRIEDA as a resource to advance geographic intelligence in LVLMs.

**Strengths:**

The paper presents a well-motivated and timely contribution that fills an underexplored gap in multimodal reasoning research. Its originality lies in extending the evaluation of vision-language models to the domain of cartographic reasoning, which requires understanding spatial relations, map symbology, scales, and orientation—skills that differ fundamentally from those assessed in standard visual question answering or chart-understanding benchmarks. The formulation of three spatial-relation categories (topological, metric, and directional) is conceptually strong and well grounded in GIS literature, offering a structured way to test spatial intelligence in LVLMs.

In terms of quality, the benchmark construction appears rigorous, combining LLM-assisted question generation with extensive human verification by multiple annotators, ensuring reliability of the ground truth. The inclusion of a human performance baseline and evaluation across both direct and contextual settings enhances the depth of analysis and provides a meaningful comparison point for model performance. The paper is also clearly written and logically organized, guiding readers from the motivation through dataset design, evaluation protocol, and results. The argumentation in the introduction—especially the rationale for curating maps from real public documents—is convincing and helps establish the benchmark’s practical relevance.

Finally, the work is significant because it pushes LVLM evaluation toward a domain that mirrors real-world map interpretation tasks relevant to areas like urban planning and environmental monitoring. By revealing a substantial human-model gap in spatial reasoning, FRIEDA sets the stage for a new line of research on geospatial and multi-map understanding in multimodal AI.

**Weaknesses:**

A few minor aspects could be clarified or expanded to further strengthen the work.

First, although the authors describe a careful validation process, the groundtruth generation pipeline (Section 3.2) partly relies on LLMs (GPT-4 and GPT-o3) to propose initial questions and reference answers before human verification. While this hybrid approach is efficient, it may introduce subtle mismatches between the automatically generated “gold answers” and human interpretations. Providing more detail on how such discrepancies were resolved or examples of discarded cases would improve confidence in the dataset’s reliability.

Second, the evaluation analysis (Section 5)—focused mainly on accuracy and error frequency—could be enriched with more nuanced insights. For example, reporting domain-wise breakdowns (e.g., geology vs. urban planning) or qualitative examples of borderline cases could illuminate specific reasoning challenges faced by LVLMs.

Finally, the contextual setting (Appendix E), although well designed, yielded similar results to the direct setting, raising questions about whether the retrieval aspect was sufficiently demanding. Clarifying how the contextual map sets were selected and randomized, or adding a more difficult retrieval scenario, could make this part of the benchmark more informative.

**Questions:**

1. Could the authors clarify in more detail how the ground-truth answers were finalized during dataset construction? Specifically, when GPT-generated “gold answers” disagreed with human annotators, how were such conflicts resolved, and were any systematic patterns of disagreement observed?

2. For the contextual setting, could the authors elaborate on the retrieval challenge? It would be helpful to understand how many candidate maps were typically presented per question and how similar or visually confounding these maps were. This clarification would help assess whether the contextual setup truly captures document-level reasoning complexity.

3. The paper mentions that the benchmark draws from multiple thematic domains (geology, environment, urban planning, etc.). Did the authors observe domain-specific variations in model performance? If not, would they consider including such an analysis to better understand where models struggle the most?

4. Could the authors comment on how they plan to ensure community adoption of the benchmark—e.g., through public leaderboards, evaluation APIs,

or standardized prompts—once anonymity is lifted

---

> ### Author Response · Authors · 2025-11-21
> **Author's Response**
>
> We thank the reviewers for the suggestions on areas where greater transparency and analysis will strengthen the paper. Below, we provide a summary of clarifications and new analyses we have added in the revision.
>
> &nbsp;
> > Could the authors clarify in more detail how the ground-truth answers were finalized during dataset construction?
>
> Our human-in-the-loop pipeline was designed to prevent LLM-driven errors from propagating into the benchmark. The LLM-proposed answers were never directly used as the final ground truth:
> - **Pre-annotation Curation**: Two question curators (one with 7 years of GIS experience and another with 2 years of experience in geospatial data) manually inspected every LLM-generated question-answer pair. They verified the gold answers against the source maps and rewrote, corrected, or discarded items if necessary.
> - **Annotator Validation**: The annotators then validated this curator-approved set.
> - **Conflict Resolution**: If the question fails to reach 2/3 agreement, the question is deemed ambiguous and is discarded. In a rare case (two questions currently in FRIEDA) where all three annotators agreed on a new answer that contradicted the curator’s answer, the item entered a secondary review. The question curator and an additional fourth reviewer re-evaluated the question. The original answer was replaced if this full review reached a consensus.
>
> This multi-stage process ensures that the final benchmark reflects human-grounded interpretation, not LLM artifacts. Regarding the discarded items, we did not observe any systematic patterns; they were a mix of textual (question) ambiguities, visually unresolvable overlaps, and rare edge cases that did not meet our standards for clarity.
>
> We have revised Section 3.2 to make the pre-annotation curation step explicit and to provide a more detailed description of our quality-control process.
>
> &nbsp;
> > For the contextual setting, could the authors elaborate on the retrieval challenge?
>
> In the contextual setting, each question is paired with multiple candidate maps, including 2–9 irrelevant maps drawn from the same source document. These irrelevant maps are thematically and visually related to the relevant map(s), making them potentially confounding for models. To prevent models from exploiting positional cues, we randomly shuffle the order of all maps in the contextual set. For further details on the selection criteria, please refer to the *Details on Contextual Setting* section of the [Response to Common Questions](https://openreview.net/forum?id=QQCadccQqU&noteId=3vSp6pqY6O).
>
> &nbsp;
> > Did the authors observe domain-specific variations in model performance?
>
> We thank the reviewer for prompting this analysis, as it clarifies the types and styles of maps where current LVLMs experience most difficulty. We have conducted a new domain-specific analysis of model performance and added Table 14 for the direct setting and Table 15 for the contextual setting (Appendix E.7)
>
> Overall, yes, we have found some domain-specific patterns in the performance. We observe that all models perform better than their average on Park & trail maps, which typically feature cleaner layouts and simpler symbology. Notably, these maps are published by the U.S. government, which means they often follow consistent, standardized cartographic conventions.
>
> In contrast, most models struggle with Geology maps and Investment maps. These categories often include highly complex legends and dense visual encodings that do not follow a uniform style. Moreover, unlike Park & Trail maps, these maps are produced by a diverse range of authors, resulting in significant stylistic variation even within a single report.
>
> &nbsp;
> > Could the authors comment on how they plan to ensure community adoption of the benchmark
>
> We are committed to making FRIEDA an active and accessible benchmark for the community. As detailed in our Reproducibility Statement, upon the end of the anonymity period we will release the complete benchmark (images, QA JSON), our full evaluation kit (including all code for data loading, inference, and evaluation), the standardized prompt (detailed in Appendix B.4), and LLM-as-Judge prompt used in our experiment (detailed in Appendix E.1).
>
> Furthermore, we plan to launch a public webpage to host the dataset and a leaderboard. This will allow the community to submit results from new models and track progress on this challenging task, thereby fostering standardized and reproducible research.

---

> > ### Comment · Reviewer_Y3WZ · 2025-11-26
> >
> > I appreciate the author's comments and they have addressed my concerns. I have updated my score.

---

> > > ### Author Response · Authors · 2025-12-03
> > >
> > > Dear Reviewer Y3WZ,
> > >
> > > As the discussion period comes to an end, we would like to thank you for your time, comments, and feedback. We are glad our responses addressed your concerns, and we appreciate the suggestions you provided, which helped improve the paper's overall quality.
> > >
> > > Thank you again for your contributions.

---

### Official Review · Reviewer_RzLv · 2025-10-31

**Soundness:** 3
**Presentation:** 3
**Contribution:** 3
**Rating:** 6
**Confidence:** 3

**Summary:**

The paper introduces FRIEDA, a new benchmark designed to test how well large vision-language models (LVLMs) perform multi-step reasoning over real maps. The benchmark spans 500 questions drawn from 17k maps across 210 documents, covering topological, metric, and directional spatial relations. Tasks require understanding legends, scales, and compass orientation, often across multiple maps. The study evaluates 11 LVLMs (Gemini-2.5-Pro, GPT-5-Think, Claude-Sonnet-4, and several open-source models). Even the best systems achieve under 40% accuracy compared to 85% human performance, revealing a major gap in cartographic reasoning.

**Strengths:**

Originality: The paper targets a truly underexplored domain. Most prior map-VQA datasets treat maps as static charts. FRIEDA is the first to combine multi-map reasoning with explicit tests of spatial relations, bridging cognitive geography and LVLM evaluation.

Quality: The benchmark is carefully built with validated questions, human agreement checks, and diverse sources. The experimental setup is solid, and the error taxonomy is informative.

Clarity: The writing is clear, structured, and well-motivated. The examples and figures make the task definition easy to grasp.

Significance: The benchmark fills an important gap between general multimodal reasoning and geospatial intelligence. The authors’ detailed error analysis provides actionable insight into how LVLMs misinterpret legends, scales, and orientation.

**Weaknesses:**

1. The dataset size (500 questions) may limit statistical robustness and fine-grained analysis.
2. While the benchmark claims multi-step reasoning, most reasoning is still visual-symbolic.
3. More discussion on relation to existing reasoning datasets like GeoChain or MapIQ would clarify complementarity and citation positioning.

**Questions:**

1. How reproducible are results across different LLM-judge configurations?
2. Did you evaluate whether models can explain their reasoning steps (not just final answers)?
3. Could GeoChain-style supervised reasoning traces help improve performance on FRIEDA?
4. Are some relation types (e.g., metric vs. topological) more prone to hallucination?

---

> ### Author Response · Authors · 2025-11-21
> **Author's Response (1/2)**
>
> We thank the reviewer for the insightful comments and for suggesting future directions in the field of cartographic reasoning. Below, we address the points raised in the review.
>
> &nbsp;
>
> > Dataset size (500 questions) may limit statistical robustness and fine-grained analysis
>
> Please refer to the *Scale of Dataset* section of our [Response to Common Questions](https://openreview.net/forum?id=QQCadccQqU&noteId=3vSp6pqY6O), which discusses why 500 carefully curated questions can yield reliable and informative evaluations.
>
> For fine-grained analyses, we assessed robustness within small subcategories using pairwise McNemar’s tests on the three proprietary models. Details on $p$-values for each subcategory and model pair are provided in Appendix D.2.
>
> Smaller subcategories can have limited statistical power. However, these analyses on subcategories are diagnostic rather than being the primary evidence supporting our claims. The key finding through FRIEDA is the consistent and substantial human-model performance gap, which remains robust across models. Subcategory analyses primarily highlight where the model struggles, even when the significance cannot be concluded due to the subcategory scale.
>
> &nbsp;
>
> > While the benchmark claims multi-step reasoning, most reasoning is still visual-symbolic.
>
> FRIEDA requires compositional visual-symbolic reasoning [1], which we refer to as multi-step cartographic reasoning. Following [2], we define multi-step reasoning as a model generating a sequence of intermediate steps to arrive at the final answer. For example, Figure 1 requires to:
> 1. Identify the neon-green region,
> 2. Use the result to determine which orange-bordered region borders it
> 3. Retrieve the name of the region.
>
> Each step depends on the previous one, forming a compositional reasoning chain rather than a single visual lookup. Thus, while FRIEDA involves visual-symbolic reasoning [3], it also explicitly tests multi-step reasoning over symbolic map elements, spatial relations, and structure.
>
> [1] Johnson, Justin, et al. "Clevr: A diagnostic dataset for compositional language and elementary visual reasoning." Proceedings of the IEEE conference on computer vision and pattern recognition. 2017
>
> [2] Paranjape, Bhargavi, et al. "Art: Automatic multi-step reasoning and tool-use for large language models." arXiv preprint arXiv:2303.09014 (2023)
>
> [3] Ji, Yuheng, et al. "Mathsticks: A benchmark for visual symbolic compositional reasoning with matchstick puzzles." arXiv preprint arXiv:2510.00483 (2025)
>
> &nbsp;
>
> > More discussion on relation to existing reasoning datasets like GeoChain or MapIQ would clarify complementarity and citation positioning.
>
> We have expanded our discussion in the Related Works (Section 6) and the Extended Related Works (Appendix G) to better position FRIEDA.
>
> As a summary:
> - *MapIQ* focuses on single-map, attribute-identification-based VQA (e.g., “What is the attribute class of [region]?”) and does not evaluate spatial relations or multi-map reasoning.
> - *GeoChain* introduces process-supervised reasoning traces, but targets geolocalization on natural images (e.g., “What language(s) are likely spoken here?") rather than spatial inference on maps.
>
> In contrast, FRIEDA is the first benchmark to integrate the full range of cartographic and GIS reasoning skills, including multi-map integration, spatial relations, legend interpretation, and multi-step comprehension. Other benchmarks address only subsets of these skills, so FRIEDA complements prior work.
>
> &nbsp;
>
> > How reproducible are results across different LLM-judge configurations?
>
> We thank the reviewer for the suggestion. We conducted additional experiments to test our LLM-judge configuration, comparing our default judge (Mistral Small 3.1) against other models and prompts. We measured agreement using Cohen’s Kappa.
> - Different Judge Model: We re-evaluated the outputs using two widely used judge models:
>    - GPT-4o: Achieved a mean Cohen’s Kappa of 0.9328 (std: 0.0739) compared to our judge.
>    - LLaMA 3.1-70B: Achieved a mean Cohen’s Kappa of 0.7904 (std: 0.2429). Evaluation disagreement mainly arose from models that do not strictly follow the ‘final answer: ‘ format, highlighting the sensitivity of judges to formatting variations.
> - Different Prompt: We also tested a different evaluation prompt from [1] using the same Mistral Small judge.
>    - Achieved a mean Cohen’s Kappa of 0.7904 (std: 0.2429). Disagreement appeared in similar areas to that of the LLaMA 3.1-70B result.
>
> Our original judge configuration shows high agreement when tested under different models and prompt settings. In addition, as noted in our reproducibility statement, we will release our exact evaluation script, model, and prompt to ensure full reproducibility.
>
> [1] Badshah, Sher, and Hassan Sajjad. "Reference-Guided Verdict: LLMs-as-Judges in Automatic Evaluation of Free-Form QA." Proceedings of the 9th Widening NLP Workshop. 2025

---

> ### Author Response · Authors · 2025-11-21
> **Author's Response (2/2)**
>
> > Did you evaluate whether models can explain their reasoning steps (not just final answers)?
>
> The models can explain their reasoning step. While our final accuracy metric focuses on the correctness of the final answer, Section 5 classifies the incorrect reasoning steps into error categories.
> We have included several qualitative examples of the reasoning traces from Gemini-2.5-Pro, GPT-5-Think, and Claude-Sonnet-4 in Appendix F.1 (Listing 1-3), illustrating their explanatory capabilities and failures.
>
> &nbsp;
>
> > Could GeoChain-style supervised reasoning traces help improve performance on FRIEDA?
>
> GeoChain-style process supervision could improve LVLM performance on FRIEDA. Guiding a model through the intermediate steps (e.g., 1. Find legend items, 2. Identify the symbol in the map, 3. Identify those satisfying the spatial relation) is a high-priority avenue for future work. We believe FRIEDA is well-positioned to serve as a challenging testbed for this purpose, as it is designed to assess compositional map-comprehension skills.
>
> &nbsp;
>
> > Are some relation types (e.g., metric vs. topological) more prone to hallucination?
>
> We found pure object hallucination (e.g., saying “Zone K” when there is no Zone K on the map) is rare, accounting for only 1.20% of errors on Gemini 2.5 Pro, with no clear correlation to a specific relation type. However, if we use a broader definition of hallucination (defined as a confident but factually incorrect assertion), then, yes, certain hallucinations are type-specific:
> - Topological relations were most prone to ‘misinterpretation of legends (25.61%)’. This can be seen as a symbolic hallucination, where the model confidently maps an abstract feature (e.g., Zone A) to the incorrect visual symbol (e.g., a blue area instead of a magenta area).
> - Distance relations were most prone to ‘incorrectly using the map scale (9.76%)’. This can be viewed as a numeric hallucination, where the model confidently performs a calculation but uses an incorrectly scaled factor.

---

> > ### Author Response · Authors · 2025-12-03
> >
> > Dear Reviewer RzLv,
> >
> > As the discussion period comes to an end, we would like to thank you for your time, comment, and feedback. We appreciate the careful evaluation of our work and the suggestions provided in your initial review. We hope we have satisfactorily answered all your questions and addressed any concerns.
> >
> > Thank you again for your contributions.

---

### Official Review · Reviewer_VsEU · 2025-10-31

**Soundness:** 2
**Presentation:** 2
**Contribution:** 2
**Rating:** 2
**Confidence:** 4

**Summary:**

This paper proposes a multi-step cartographic reasoning benchmark dataset for cartographic reasoning in 3 dimensions: topological, metric, and directional spatial relationships. The maps are sourced from publicly available dataset, 500 questions are drafted by propriety models. Evaluation was performed across 11 LVLMs, providing 5 findings.

**Strengths:**

The study does well to highlight the gap in research across all the other Map VQA studies. For example, it highlights that multi-map VQA is an area not evaluated, and if it is evaluated, it’s very limited (Kazemi et al, 2025).

The study does a good job linking the human cognition aspect of map querying and exploring how it would be interesting to see how LVLMs reason over maps. A strong real world foundation is provided in this study and it drives the need for the benchmark presented.

Evaluation metrics are clear and fitting for each category of questions tested. LLM as a judge is used for textual questions while distance based answers are evaluated on MAPE. Directional answers are mapped to the relevant locations labeled with North, South, East, West metrics and all directions in between.

Although the queries are drafted by proprietary models, they are manually reviewed by humans to maintain the quality. Provides logical findings through extensive evaluations and analyses, found a huge gap between humans and models. Also, the overall writing is good and easy to follow.

**Weaknesses:**

The motivation of this paper is weak. I am not convinced that cartographic QA is different from the other MapQAs, even after reading the whole body of Section 2. Further comparison between related works should be highlighted to demonstrate the novelty of this research direction.

The scale of the benchmark set is moderately small (500 questions), compared to the other LVLM benchmarks. A very limited novelty is there. Doesn't compare with vast literature of prior work on MapQA[1], MapWise[2], MapIQ [3], MapBench[4], CartoMark[5], MapQA (GQA)[6] and many more. A proper comparison should be there with prior works to show how this is different.

The task definition introduces too many dimensions for each category. In Spatial Reasoning, many dimensions to the category are formalized, giving the impression that the benchmark would extensively test against it. Figure 2 and Table 1 don’t align with the in-depth explanation for the tasks formalized in section 2, so there is a disconnect with how each dimension is tested.

In the benchmark section, 500 questions are evaluated against 17,030 maps from 210 documents. Because it’s a multi-map task, the map-per-question distribution may be uneven; while the authors note that each question uses “two or more maps,” they don’t state a maximum. That matters: LVLM performance can differ wildly—two-map queries are manageable for state-of-the-art models, but queries involving hundreds of maps will strain context windows (and humans).

Limiting the scope to Latin-script text is concerning, and demonstration in Figure 1 does not look like it is from Latin-script text. Also, does it mean that all components including legends, region descriptions are translated or retrieved from references?

References [1] https://arxiv.org/abs/2211.08545 [2] https://arxiv.org/pdf/2409.00255v1 [3] https://arxiv.org/abs/2507.11625 [4] https://arxiv.org/pdf/2503.14607 [5] https://www.nature.com/articles/s41597-024-04057-7 [6] https://arxiv.org/pdf/2503.07871

**Questions:**

I am skeptical towards accepting adjacent labels. Also, isn’t it ambiguous to include partially-agreed questions in the benchmark dataset? Does the findings change if we do not consider those partial-agree items?

How did the authors analyze the errors and misinterpretations of Gemini-Pro? Are those manually checked following the reasoning trace?

In the “Performance by spatial relation” paragraph, are there any reasons why the models are ranked as in Fig 4? Does the amount or ratio of spatial understanding data in each train split matters?

How do RL-trained models perform on this benchmark? Are there any inductive bias observed from visual encoders? Are there any underlying biases or trends from the training set? These points are revealed that these two points matter [1,2].

Will there be an explanation for the distribution of questions for multi-map querying i.e how many maps does one question have?

References
[1] Wang et al. VideoHallucer: Evaluating Intrinsic and Extrinsic Hallucinations in Large Video-Language Models. arxiv:2406.16338
[2] Li et al. Vidhalluc: Evaluating Temporal Hallucinations in Multimodal Large Language Models for Video Understanding. CVPR 2025.

---

> ### Author Response · Authors · 2025-11-21
> **Author's Response (1/2)**
>
> We thank the reviewer for their constructive feedback and the opportunity to clarify FRIEDA’s core contribution and its distinction from prior work. Below, we address the comments, highlighting our motivation, novelty, and new analyses incorporated in the revised manuscript.
>
> &nbsp;
> > The scale of the benchmark set is moderately small
>
> > Further comparison between related works
>
> Please refer to *Scale of Dataset* of [Response to Common Questions](https://openreview.net/forum?id=QQCadccQqU&noteId=3vSp6pqY6O) for our discussion on dataset size.
>
> Our motivation for curating FRIEDA is that existing map QA benchmarks do not fully capture the core skills required for human-like map comprehension. Many prior benchmarks test isolated or simplified abilities, while FRIEDA reflects the set of skills humans incorporate when interpreting maps. Specifically, FRIEDA evaluates questions that require models to:
> - Understand spatial relationships (topological, metric, directional)
> - Be robust to heterogeneous map styles (being able to interpret common elements such as legend, scale, and compass regardless of stylistic variations)
> - Integrate information across multiple maps
> - Identify and retrieve relevant maps necessary for answering a query.
>
> Appendix G (Table 17) compares FRIEDA with nine other map QA benchmarks, including MapQA (Chang et al., 2022), MapWise, MapIQ, MapBench, and CartoMark, demonstrating that FRIEDA is the only dataset that can comprehensively evaluate all of these aspects. To better position FRIEDA within the broader map QA literature, we have also expanded Related Works (Section 6) and added an Extended Related Works (Appendix G), highlighting its complementary role and unique contribution to the field.
>
> [We do not compare our work with MapQA (Li et al., 2025), as it is not a visual benchmark. FRIEDA requires models to extract entity location from the map image itself, and in many cases, the entities are not present in standard map databases, demanding accurate visual-spatial understanding.]
>
> &nbsp;
>
> > Task definition dimensions
>
> We thank the reviewer for the opportunity to elaborate on the task definition of FRIEDA and clarify how its design reflects the core skills required for human-like map comprehension. Our formalization enumerates a broad set of dimensions to provide a rigorous taxonomy of the map-comprehension task. This level of detail may give the impression that all dimensions are evaluated in parallel. In practice, FRIEDA applies these dimensions hierarchically:
> - Spatial relations (Objectives): Core objective of each question; analyses in the main text (Table 2)
> - Map elements (Tools): Visual/symbolic components needed to solve the objective; analyses in Appendix E.6 (Table 12-13)
> - Multi-map & Contextual Setting (Scope): Determines the amount of information integration and retrieval required; analyses in Appendix E.5; E.3 (Table 10-11; 8)
>
> All questions are fully annotated across these dimensions, enabling fine-grained analysis beyond the main text. To improve alignment, Table 1 now summarizes map-element usage and scope, while Figure 2 focuses on spatial relations. We also added a Sankey diagram (Appendix C - Figure 11) that visualizes the hierarchical structure from map counts to spatial relations to map elements.
>
> &nbsp;
>
> > Will there be an explanation for the distribution of questions for multi-map querying i.e how many maps does one question have?
>
> Of the 298 multi-map questions, 295 require integrating information from two maps, 2 require three maps, and 1 requires four maps.
>
> We have added this information to our updated revision (Section 3.1)
>
> &nbsp;
> > Limiting the scope to Latin-script text
>
> We want to clarify that ‘Latin-script’ refers to documents using Latin characters (ISO standard), including English, Spanish, and French. No translation was performed; all legends, titles, and text are retained in the original language (including Figure 1).
> We have updated the terminology to ‘Latin characters’ for clarity.

---

> ### Author Response · Authors · 2025-11-21
> **Author's Response (2/2)**
>
> > Accepting adjacent labels
>
> We chose to mark responses as correct if they were within one adjacent label, as directional answers are often perceptual. For example, even with humans, one may label a 20-degree angle as ‘North East’ while another might label it as ‘East’. The ambiguity is further amplified in directional questions involving a large polygon and a point, where the perceived direction can shift based on the polygon’s shape or centroid.
>
> To address the reviewer’s concern, we also recalculated performance on the 83 directional answer-type questions using a strict Exact Match metric (i.e., accept only if the direction exactly matches). The results for the direct setting are as follows:
>
> | Model | Adjacent Score | Exact Match Score | Percent Change |
> | --- | --- | --- | --- |
> | Human Average | 92.15 | 72.69 | 19.46 |
> | Gemini 2.5 Pro | 73.49 | 50.60 | 22.89 |
> | GPT-5-Think| 72.29 | 53.01 | 19.28 |
> | Claude Sonnet 4 | 59.04 | 31.33 | 27.71 |
> | LLaVA-NeXT-110B | 0.00 | 0.00 | 0.00 |
> | GLM-4.5V-108B | 1.23 | 0.00 | 1.23 |
> | InternVL3-78B | 36.14 | 13.25 | 22.89 |
> | LLaVA-OneVision-72B | 30.12 | 10.84 | 19.28 |
> | Qwen2.5-VL-72B | 54.22 | 32.53 | 21.69 |
> | InternVL3.5-38B | 38.55 | 13.25 | 25.30 |
> | Ovis2-34B | 0.00 | 0.00 | 0.00 |
> | Ovis2.5-9B-Think | 53.01 | 4.82 | 48.19 |
>
> As expected, performance decreases under the strict Exact Match evaluation for all models. However, the overall pattern remains consistent: the human-model gap is substantial, and proprietary models generally outperform open-source models.
>
> &nbsp;
>
> > Isn’t it ambiguous to include partially-agreed questions in the benchmark dataset? Does the findings change if we do not consider those partial-agree items?
>
> The Partial-agree questions indicate cases where annotators did not provide the correct answer. Each question has a well-defined gold answer, so the questions themselves are not ambiguous. Partial agreement reflects human error rather than unclear or obscure phrasing.
>
> The overall trend and findings remain consistent even when we only analyze the All-agree subset (297 questions):
>
> | Model | Full Score | All-Agree Score (297) |
> | --- | --- | --- |
> | Human Average | 84.87 | 93.93|
> | Gemini 2.5 Pro | 38.20 | 46.13 |
> | GPT-5-Think| 37.20 | 44.11 |
> | Claude Sonnet 4 | 31.60 | 37.04 |
> | LLaVA-NeXT-110B | 8.60 | 9.43 |
> | GLM-4.5V-108B | 6.40 | 8.67 |
> | InternVL3-78B | 11.00 | 13.80 |
> | LLaVA-OneVision-72B | 13.00 | 14.48 |
> | Qwen2.5-VL-72B | 25.60 | 28.28 |
> | InternVL3.5-38B | 14.20 | 14.81 |
> | Ovis2-34B | 17.80 | 20.54 |
> | Ovis2.5-9B-Think | 25.80 | 29.97 |
>
> The result confirms that the difficulty of FRIEDA stems from the model’s limitation in cartographic reasoning, not from noise introduced by Partial-agree questions. We have added these results to Table 9 in Appendix E.4.
>
> &nbsp;
>
> > How did the authors analyze the errors and misinterpretations of Gemini-Pro? Are those manually checked following the reasoning trace?
>
> Yes, our error analysis on Gemini2.5 Pro was performed by manually checking the reasoning traces to the incorrect questions.
>
> &nbsp;
>
> > Are there any reasons why the models are ranked as in Fig 4?
>
> The order of models in the legend of Figure 4 was previously arbitrary. To reduce confusion, we have reordered the legend so that proprietary models are listed from highest to lowest average accuracy. We thank the reviewer for pointing this out.
>
> &nbsp;
>
> > Does the amount or ratio of spatial understanding data in each train split matters?
>
> All of our experimentation is conducted in a zero-shot manner, and, therefore, we do not have a train split.
>
> &nbsp;
>
> > How do RL-trained models perform on this benchmark?
>
> We found that RL-trained models (e.g., GLM-4.5V with RLCS, InternVL-3.5 with MPO/GSPO, Ovis2.5 with DPO/GRPO) generally outperform pure SFT models (e.g., LLaVA-NeXT and LLaVA-OneVision). This suggests that RL-tuning does help, but is not nearly enough to solve cartographic reasoning.
>
> &nbsp;
>
> > Are there any inductive bias observed from visual encoders? Are there any underlying biases or trends from the training set?
>
> Yes, there are observable inductive biases arising from the choice of visual encoder and the pre-training data. Models that rely on CLIP or SigLIP (e.g., LLaVA-NeXT, LLaVA-OneVision) consistently perform worse on FRIEDA. These might be artifacts due to the CLIP training objective being global image-text alignment rather than localized reasoning. In contrast, models that use ViT or NaViT (e.g., Ovis2, Ovis2.5) perform noticeably better, suggesting that encoders that preserve high-resolution spatial detail offer strong performance on FRIEDA-style spatial reasoning tasks.
> InternViT (i.e., InternVL3 and InternVL3.5), which is trained for object detection and segmentation, outperforms CLIP-based models. These patterns indicate that visual encoders optimized for fine-grained spatial structure, rather than global alignment, better support types of spatial relations evaluated in FRIEDA.

---

> > ### Comment · Reviewer_VsEU · 2025-11-26
> > **Thanks for the Response**
> >
> > Thank you for your responses and clarifications. After considering your rebuttal, I have decided to improve my score for the submission.

---

> > > ### Author Response · Authors · 2025-12-03
> > >
> > > Dear Reviewer VsEU,
> > >
> > > As the discussion period comes to an end, we would like to thank you for dedicating your time to evaluating our work and providing constructive feedback. Your comments have been valuable in helping us improve the clarity and quality of the paper, and we are glad our response addressed your concern.
> > >
> > > Thank you again for your contribution.

---

### Official Review · Reviewer_8fK6 · 2025-11-07

**Soundness:** 3
**Presentation:** 4
**Contribution:** 3
**Rating:** 6
**Confidence:** 5

**Summary:**

This paper introduces FRIEDA, a benchmark designed to evaluate multi-step cartographic reasoning in vision-language models (VLMs). FRIEDA focuses on reasoning over real-world maps and includes both single-map and multi-map questions.
The benchmark contains 500 questions and supports two evaluation modes: (1) the direct setting, where the relevant map(s) are provided, and (2) the contextual setting, where models must first identify the relevant map(s) from a document set before reasoning.

The authors evaluated 11 VLMs, finding that even the strongest models achieve less than 40% accuracy, far below the human baseline of 84.9%.

**Strengths:**

1.The paper addresses the need for a benchmark that evaluates VLMs’ reasoning abilities over diverse maps. In addition to direct map-based question answering, it includes a retrieval-like setting, where the model must first identify useful maps before answering.

2.All questions are verified by human annotators, ensuring high dataset quality. The appendix provides clear documentation of the annotation and curation processes, which helps readers understand how the dataset was built.

3.The paper is well-written and easy to follow.

**Weaknesses:**

1. With only 500 questions, the dataset may be too small for robust model evaluation, especially when analyzing performance across subcategories (e.g., Table 2 and Figure 4). Can you conduct significance tests on the small categories?

2. The paper asserts that all questions require visual reasoning, but no text-only baseline (question without the map) is reported. This makes it unclear whether VLMs rely on internal knowledge to answer some questions.

3. The questions are synthesized by LLMs given maps, which may make them too artificial and not well aligned with how humans naturally seek information from maps. The textual context of the original documents could better reflect real human interest in these maps but is not utilized. Beyond verifying the validity of these questions, human annotators should also analyze whether the questions resemble those naturally encountered in real-world settings.

4. It is unclear whether the questions contain explicit spatial references (e.g., “the red dots in the image”) or rely only on abstract factual references such as legend words. Prior work [1] suggests that VLMs struggle with understanding visual features, which is particularly important in map interpretation.

5. The contextual setting experiment setup is not clearly explained.

In general, I believe this paper is well motivated and well executed, though it still has a few concerns that need to be addressed. I would be happy to adjust my scores after seeing the authors’ response.

[1] Rahmanzadehgervi, Pooyan, et al. "Vision language models are blind." Proceedings of the Asian Conference on Computer Vision. 2024.

**Questions:**

Besides the points listed under Weaknesses, I have a few clarification questions.

1. For the multi-map questions, how many maps are provided on average in the direct and contextual settings?

2. In the contextual setting, do the VLMs see the entire document or only the map images when answering a question?

3. How does the contextual setting compare in difficulty for humans? Are these questions also easy for human annotators?

---

> ### Author Response · Authors · 2025-11-21
> **Author's Response (1/2)**
>
> We thank the reviewer for pointing out potential areas that can strengthen the value of FRIEDA, as well as points that require clarification. Below are responses to the questions and concerns raised in the review.
>
> &nbsp;
>
> > With only 500 questions, the dataset may be too small for robust model evaluation, especially when analyzing performance across subcategories. Can you conduct significance tests on the small categories?
>
> We would like to thank the reviewer for the suggestion. We conducted pairwise McNemar’s tests to assess robustness within small subcategories. For subcategories with fewer than 50 disagreement counts, we used exact binomial test; for larger counts, we used $chi$-squared test with correction. These details have been added to Appendix D.2.
>
> The results show that some differences remain statistically significant under these tests. To highlight few results:
> - Multi-map questions (298 questions): GPT-5-Think outperforms Sonnet-4 ($p < 0.01$) and shows borderline significance against Gemini 2.5 Pro ($p \approx 0.05$).
> - Equal (54 questions): Although GPT-5-Think achieves higher raw correct count (24 v. 18), the effect sizes are not enough to reach statistical significance ($p \approx 0.10$).
>
> Some subcategories have limited statistical power due to small sample sizes; however, these analyses are diagnostic rather than the primary basis for our claims. The central finding we aim to highlight is the consistent and substantial human-model performance gap. Subcategory analyses illustrate where models potentially struggle, even when significance is inconclusive at a very small scale.
>
> Regarding the overall benchmark size please refer to *Scale of Dataset* of the [Response to Common Questions](https://openreview.net/forum?id=QQCadccQqU&noteId=3vSp6pqY6O).
>
> &nbsp;
>
> > no text-only baseline (question without the map) is reported
>
> Our curation process was explicitly designed to create questions that cannot be answered without the images. In addition to screening each question for searchability by using GPT with web search enabled, we excluded questions that can be answered without inspecting the map images.
>
> To empirically validate this and quantify any potential knowledge leakage, we ran the ‘text-only’ baseline experiment suggested by the reviewer. We evaluated the best proprietary model (Gemini 2.5 Pro) and the best open-source model (Ovis 2.5-9B-Think) on all 500 questions, providing only the question text. The results confirm our design:
> - Gemini 2.5 Pro accuracy dropped from 38.20% (with image) to 9.0% (without image)
> - Ovis2.5-9B-Think accuracy dropped from 25.80% (with image) to 4.0% (without image)
>
> We analyzed the few remaining correct answers and found that they occurred on questions where a correct response can be accidentally obtained through random guessing (e.g., ‘how many points exists’, ‘what is the orientation’).
>
> &nbsp;
>
> > The questions are synthesized by LLMs given maps, which may make them too artificial and not well aligned with how humans naturally seek information from maps.
>
> While we used LLMs to propose candidate questions, all questions were subsequently reviewed by two human curators (one with 7 years of GIS experience and another with 2 years of experience in geospatial data) to ensure they were logical, unambiguous, and plausible.
>
> Although mining questions directly from document text is valuable, FRIEDA’s primary goal is to benchmark cartographic reasoning. These skills (e.g., spatial relation, legend interpretation, multi-map integration) are often implicit and not explicitly articulated in a document’s text. Our LLM-assisted generation, combined with expert curation, enables us to target these complex reasoning requirements systematically. We believe a model that performs well on FRIEDA will, by design, be able to answer the natural questions humans seek from maps.
>
> &nbsp;
>
> > It is unclear whether the questions contain explicit spatial references (e.g., “the red dots in the image”) or rely only on abstract factual references such as legend words.
>
> FRIEDA questions use abstract factual references (e.g., legend words); the core goal of FRIEDA is to assess whether models can connect abstract textual cues to corresponding spatial/visual references (e.g., a red dot), mirroring how humans interpret maps.
>
> Our error analysis shows that the primary failure mode of LVLMs is ‘misinterpretation of legend’, where models incorrectly link a legend term to the corresponding map symbol. This aligns with the findings in [1], which report that VLMs often struggle to understand visual references.
>
> [1] Rahmanzadehgervi, Pooyan, et al. "Vision language models are blind." Proceedings of the Asian Conference on Computer Vision. 2024.

---

> ### Author Response · Authors · 2025-11-21
> **Author's Response (2/2)**
>
> > The contextual setting experiment setup is not clearly explained
>
> > In the contextual setting, do the VLMs see the entire document or only the map images when answering a question?
>
> In the contextual setting, the model only sees the map images when answering questions. We provide a detailed explanation of the contextual setting and rationale behind constructing the contextual set in the *Details on Contextual Setting* section of the [Response to Common Questions](https://openreview.net/forum?id=QQCadccQqU&noteId=3vSp6pqY6O).
>
> &nbsp;
>
> > For the multi-map questions, how many maps are provided on average in the direct and contextual settings?
>
> The statistics of the 298 multi-map questions are as follows:
> - FRIEDA-Direct: 295 requires integrating information from two maps; 2 require three maps; and 1 requires four maps.
> - FRIEDA-Contextual: For each question, we provide 4-10 maps, including the ones required to answer it. Each question has an average of 9.55 map, with a standard deviation of 1.25.
>
> &nbsp;
>
> > How does the contextual setting compare in difficulty for humans? Are these questions also easy for human annotators?
>
> During human annotation, we validated questions only in the direct setting, where annotators achieved 84.87% accuracy. In Section 5, we found 88.03% agreement in model performance across questions in direct and contextual settings, indicating that retrieval is not the primary bottleneck even for the LVLMs. Since selecting the correct map from a small, thematically related set, using visual cues such as map title and content, is trivial for humans, we expect human accuracy in the contextual setting to be nearly identical to the 84.87% observed in the direct setting.

---

> > ### Comment · Reviewer_8fK6 · 2025-11-26
> >
> > I appreciate the authors’ response which addressed my questions well. I maintain my original recommendation.

---

> > > ### Author Response · Authors · 2025-12-03
> > >
> > > Dear Reviewer 8fK6,
> > >
> > > As the discussion period comes to an end, we would like to thank you for your time, comments, and feedback. We appreciate the careful evaluation of our work and insights provided by the review. Your suggestions have been valuable in helping us improve the quality of the paper.
> > >
> > > Thank you again for your contribution.

---

### Author Response · Authors · 2025-11-21
**Response to Common Questions**

We thank all the reviewers for the detailed reviews. We have revised the manuscript to reflect suggestions, and all changes are marked in purple. Below, we address the common questions raised across the reviews.

&nbsp;
### **Scale of Dataset**

While FRIEDA consists of 500 questions, its value lies in its depth and authenticity. Map comprehension is not a simple visual recognition task but a multimodal comprehension skill that humans develop over time. To capture this, we grounded our question design in established taxonomies from GIS and educational geography [1].

Existing map VQA benchmarks provide a valuable foundation, yet they assess isolated or simplified abilities. As a result, they do not capture the breadth of skills required for genuine map comprehension. FRIEDA directly addresses this gap by ensuring that each question necessitates combining distinct skills (e.g., scale calculation, legend interpretation, spatial relation analysis) to arrive at the answer. This distinction is reflected in our human performance baseline (\~84%), which contrasts with the near-perfect scores (\~95%) reported in prior datasets. This contrast does not indicate that FRIEDA’s questions are obscure; instead, it demonstrates that FRIEDA targets the deliberate and integrative thinking process of real-world map reading, which narrower benchmarks do not. Our experiments show that the 500 curated questions provide reliable and informative evaluations across models, as evidenced by consistent human-model performance gaps.

[1] Michael F. Goodchild. The fourth r? rethinking gis education. ArcUser Fall, pp. 46–51, 2012

&nbsp;
### **Details on Multi-map Setting**

Our dataset consists of 298 multi-map questions. Among these, 295 require integrating information from two maps, 2 require three maps, and 1 requires four maps.

We have added these statistics to Section 3.1 of the manuscript.

&nbsp;
### **Details on Contextual Setting**

In the contextual setting, we provide the model with the question, the relevant maps, and a set of additional maps that are not relevant to the query (irrelevant maps). The model receives only the map images and the question text; it does not see the source document or any surrounding textual context. The irrelevant maps are drawn from the same source document as the relevant map and are selected based on page proximity. We assume that maps from the same chapter or nearby pages within a report typically share a thematic purpose and exhibit related visual or conceptual content. This design reflects realistic map-reading scenarios in which multiple maps with related visual styles co-occur, and the reader must identify the appropriate one. Each question includes 2-9 irrelevant maps, with an average relevant-to-irrelevant map ratio of 1:5.71.

We have revised Section 3.1 for the statistics and Section 3.2 to provide a clear explanation of the setting.

---

### Meta-Review · Area_Chair_gmBj · 2025-12-25

**Summary:**

Reviewers generally agreed that the paper introduces a well-motivated and carefully curated benchmark for multi-step cartographic reasoning in LVLMs, addressing an underexplored but important gap. Strengths consistently highlighted include high-quality data curation, clear task formulation grounded in GIS and cognitive geography, and a large human–model performance gap demonstrating the difficulty of the task. The main concerns centered on the moderate dataset size, clarity and rigor of the evaluation (e.g., statistical significance, baselines), positioning relative to prior map QA benchmarks, and clarity of the contextual setting and task definitions.

**Reviewer Concerns:**

**Addressed by the rebuttal:**

Dataset scale and statistical robustness: Additional significance tests (e.g., McNemar’s tests) and clarifications on the intended diagnostic role of subcategory analyses.

Lack of text-only baseline: New experiments confirmed strong performance drops without visual input.

LLM-generated questions and reliability: Detailed human-in-the-loop curation and conflict-resolution process clarified.

Contextual and multi-map setting clarity: Explicit statistics and clearer experimental descriptions were provided.

Relation to prior work: Expanded related-work discussion and comparison tables improved positioning.

Evaluation robustness and reproducibility: Additional LLM-judge analyses and commitments to release code and data.

**Still outstanding:**

Dataset size limitations remain an inherent constraint, especially for fine-grained analyses.

Novelty relative to existing MapQA benchmarks remains debated by at least one reviewer.

Depth of evaluation analyses (e.g., broader domain-wise insights or more challenging retrieval settings) could be further strengthened.

**Reviewer Scores:**

Reviewer 8fK6: Likely unchanged (maintains marginal accept / borderline stance).

Reviewer VsEU: Improved, moving from reject toward a more favorable score after rebuttal.

Reviewer RzLv: Likely unchanged (maintains marginal accept with remaining reservations).

Reviewer Y3WZ: Improved, updating to a clearer accept after concerns were addressed.

---

### Decision · Program_Chairs · 2026-01-26

Accept (Poster)